# The synthetic NLR RGA5$^{HMA5}$ requires multiple interfaces within and outside the integrated domain for effector recognition

Xin Zhang [1,2,5], Yang Liu [1,3,5], Guixin Yuan[1,2], Shiwei Wang[1,2], Dongli Wang[1,3], Tongtong Zhu[1,3], Xuefeng Wu[1,3], Mengqi Ma[1,3], Liwei Guo[1,4], Hailong Guo[1], Vijai Bhadauria [1], Junfeng Liu [1,3] ✉ & You-Liang Peng [1,2] ✉

Some plant sensor nucleotide-binding leucine-rich repeat (NLR) receptors detect pathogen effectors through their integrated domains (IDs). Rice RGA5 sensor NLR recognizes its corresponding effectors AVR-Pia and AVR1-CO39 from the blast fungus *Magnaporthe oryzae* through direct binding to its heavy metal-associated (HMA) ID to trigger the RGA4 helper NLR-dependent resistance in rice. Here, we report a mutant of RGA5 named RGA5$^{HMA5}$ that confers complete resistance in transgenic rice plants to the *M. oryzae* strains expressing the noncorresponding effector AVR-PikD. RGA5$^{HMA5}$ carries three engineered interfaces, two of which lie in the HMA ID and the other in the C-terminal Lys-rich stretch tailing the ID. However, RGA5 variants having one or two of the three interfaces, including replacing all the Lys residues with Glu residues in the Lys-rich stretch, failed to activate RGA4-dependent cell death of rice protoplasts. Altogether, this work demonstrates that sensor NLRs require a concerted action of multiple surfaces within and outside the IDs to both recognize effectors and activate helper NLR-mediated resistance, and has implications in structure-guided designing of sensor NLRs.

Many crop diseases, including rice blast caused by *Magnaporthe oryzae*, pose a serious threat to global crop production and food security[1,2]. To cause such diseases, pathogens secrete a diverse array of effectors into host cells to subdue plant immunity[3]. To counter the effectors, plants have evolved nucleotide-binding leucine-rich repeat (NLR) immune receptors that recognize the avirulence (Avr) effectors either directly through physical binding or indirectly by monitoring the effector-mediated modification of guardee or decoy proteins and activate downstream immune responses[4–6]. Some of these NLRs are genetically linked and physically paired, one of which functions as a sensor receptor and the other as a helper NLR[7–11]. Both sensor and helper NLRs share a tripartite domain architecture: an N-terminal coiled-coil (CC) or Toll/Interleukin-1 receptor domain, a central nucleotide-binding (NB-ARC) domain, and a C-terminal leucine-rich repeat (LRR) domain[7–11]. However, sensor NLRs in the paired NLR systems usually carry an additional non-canonical integrated domain (ID; e.g., heavy metal-associated [HMA] or WRKY domains) that serves as "bait" to entice pathogen effectors[7–15]. Therefore, these IDs act as excellent targets for molecular engineering to create novel sensor NLR receptors.

Several studies have recently reported synthetic sensor NLRs with extended or altered effector recognition specificities via molecular engineering of the HMA IDs of Pik1 and RGA5[16–18]. RGA5/RGA4 and Pik-1/Pik-2 are two paired NLR receptors in rice that confer blast resistance,

¹The State Key Laboratory of Maize Bio-breeding, Joint International Research Laboratory of Crop Molecular Breeding, Ministry of Agriculture Key Laboratory for Crop Pest Monitoring and Green Control, College of Plant Protection, China Agricultural University, 100193 Beijing, China. ²Frontiers Science Center for Molecular Design Breeding, China Agricultural University, 100193 Beijing, China. ³Joint International Research Laboratory of Crop Molecular Breeding, China Agricultural University, 100193 Beijing, China. ⁴State Key Laboratory for Conservation and Utilization of Bio-Resources in Yunnan, Yunnan Agricultural University, 650201 Kunming, China. ⁵These authors contributed equally: Xin Zhang, Yang Liu. ✉e-mail: jliu@cau.edu.cn; pengyl@cau.edu.cn

within which RGA5 and Pik-1 function as the sensor NLRs, and RGA4 and Pik-2 as the helper NLRs. Notably, both RGA5 and Pik-1 carry an HMA ID for recognizing their corresponding *Magnaporthe* **A**vrs and To**x**B-like (MAX) effectors from *M. oryzae*[9,10,12–15]. However, the two NLR systems differ in their working mechanisms. Pik-1 binds to its corresponding MAX effector AVR-PikD via an interface comprising the β2-β3-β4 sheet in the HMA ID while RGA5 physically interacts with the MAX effectors AVR-Pia and AVR1-CO39 through an interface composed of the α1 helix and β2 strand in the HMA ID[9,10,13,14]. Rice germplasm contains multiple alleles of *Pik1*, which possess distinct effector recognition specificities and are polymorphic, mainly in their HMA IDs. For instance, Pikp-1 only recognizes AVR-PikD, while Pikm-1 can perceive AVR–PikD and two additional AVR-Pik variants (AVR-PikA and AVR-PikE)[15]. An engineered Pikp1 carrying the AVR-Pik binding interface of Pikm-1-HMA gained the capacity of Pikm1 to recognize the AVR-Pik variants, phenomimicking Pikm-1-mediated cell death in *N. bethamiana*[16]. By engineering the AVR1-CO39 interface of RGA5, we recently generated a designer NLR receptor RGA5[HMA2] that confers specific resistance in transgenic rice to the *M. oryzae* strains expressing AVR-Pib[17]. However, Cesari et al. recently reported that the RGA5 mutants carrying the AVR-PikD binding interface of Pikp1-HMA can interact with the noncorresponding effector AVR-PikD but are unable to confer rice resistance to the *M. oryzae* strains expressing AVR-PikD[18]. Further, Wang et al. generated an RRS1 variant by introducing the *Phytoplasma* effector SAP05-dependent degron domain to the C-terminus of RRS1. This synthetic RRS1 receptor can recognize SAP05 but is unable to confer full resistance in transgenic Arabidopsis against the *Phytoplasma*[19]. These studies raise a key question as to why de novo effector-binding results in different outcomes in NLR activation in the host plants. We reason that de novo effector-binding per se is necessary but insufficient for designer sensor NLRs to trigger full immune responses in their host plants. Therefore, this study set out to define interfaces in the RGA5 HMA ID and its adjacent region that are required for designer sensor NLRs to activate RGA4-dependent immunity in rice.

In this study, we contrived four RGA5-HMA mutants based on the crystal structures of AVR-PikD, the RGA5-HMA/AVR1-CO39 complex, and the Pikp-HMA/AVR-PikD complex. Intriguingly, all the mutants gained binding affinity to AVR-PikD, but only RGA5[HMA5] harboring the RGA5-HMA5 ID, activated RGA4-dependent cell death of rice protoplasts and conferred resistance in transgenic rice plants to the *M. oryzae* strains expressing AVR-PikD. We identified three interfaces bound by AVR-PikD in the C-terminus of RGA5[HMA5], two of which are located within the HMA ID and the other one in a Lys-rich stretch tailing the HMA ID. Notably, the RGA5 mutants having one or two of the three engineered interfaces failed to activate RGA4-dependent cell death in rice protoplasts. Further, the positively charged Lys residues in the Lys-rich stretch interface are essential to the derepression of RGA4 by RGA5[HMA5]. Altogether, this study demonstrates that synthetic sensor NLRs require concerted action of multiple interfaces within and outside IDs for the effector-binding and receptor-activation functions, and has implications for structure-guided rational designing of sensor NLR receptors. This study also represents a significant advance towards designing NLR receptors with distinct specificity of recognition, which can be deployed for breeding multiline cultivars to help prevent the erosion of host resistance[20].

## Results

### Designer NLR receptors carrying a single engineered effector-binding interface within the RGA5-HMA bind to the non-corresponding MAX effector AVR-PikD but fail to trigger RGA4-dependent cell death in rice protoplasts

Previous studies have revealed that the Pik1-HMA and the RGA5-HMA domains are structurally similar, comprising a four-stranded antiparallel β-sheet and two α-helices packed in an α/β sandwich model[10,13].

RGA5-HMA domain interacts mainly with the β2 strand of AVR1-CO39 via the α1 helix-β2 strand interface[13] (Fig. 1a). By engineering this interface along with the Lys-rich stretch located immediately after the HMA domain, we generated a designer NLR receptor RGA5[HMA2] that confers resistance in transgenic rice plants to the *M. oryzae* strains expressing the noncorresponding MAX effector AVR-Pib[17]. Therefore, we reasoned whether this interface could be resurfaced to generate an RGA5-HMA mutant capable of binding to another noncorresponding effector AVR-PikD. Structural superimposition of AVR-PikD (PDB ID:5A6W) with the complex of AVR1-CO39/RGA5-HMA (PDB ID:5ZNG) suggested that the M1016V mutation in RGA5-HMA may form hydrophobic interactions with A67 and G68 in AVR-PikD, and reduces steric hindrance at the interface between RGA5-HMA and AVR-PikD (Fig. 1a). The G1009D and S1027V mutations were adopted to block the interaction with AVR1-CO39 based on the previously contrived designer NLR RGA5[HMA2] [17]. We thus generated a mutant of RGA5-HMA carrying the G1009D, M1016V, and S1027V mutations, named RGA5-HMA3. A previous study reported that the β2-β3-β4 sheet within Pik-HMA is the interface for interaction with AVR-Pik[16]. E230 in the β3 strand of Pikp-HMA is one of the key residues interacting with H46 of AVR-PikD or N46 of other AVR-Pik effectors by a salt bridge or hydrogen bond, which corresponds to V1039 in RGA5-HMA that may decrease the interaction with AVR-PikD (Fig. 1b, d)[10,21]. Further comparison of the interfaces in the structures of RGA5-HMA, Pikp-HMA, Pikm-HMA, and the complexes of Pik-HMA bound to different AVR-Pik effectors suggested that E1070 immediately after the β4 strand of RGA5-HMA corresponds to the N261 residue of Pikp-HMA, which caused the "looping out" of this region of Pikp1-HMA, thereby decreasing the binding affinity as compared with Pikm1-HMA that lacks the N residue (Fig. 1c). We then generated another RGA5-HMA variant carrying the V1039E substitution and the E1070 deletion, named RGA5-HMA4, for recognizing AVR-PikD (Fig. 1d).

To test whether RGA5-HMA3 and RGA5-HMA4 are able to bind to AVR-PikD, we first performed the yeast two-hybrid (Y2H) assay. As shown in Fig. 2a, both RGA5-HMA3 and RGA5-HMA4 interacted with AVR-PikD in the yeast cells. To further confirm the interactions in plant cells, we replaced the wild-type RGA5-HMA with RGA5-HMA3 and RGA5-HMA4, generating two designer NLRs called RGA5[HMA3] and RGA5[HMA4]. Co-IP assays showed that when coexpressed in *N. benthamiana*, both the HA-tagged RGA5[HMA3] and RGA5[HMA4] were coimmunoprecipitated with GFP-tagged AVR-PikD. In contrast, RGA5 was not coimmunoprecipitated with AVR-PikD (Fig. 2b). These results indicated that RGA5[HMA3] and RGA5[HMA4] could bind to the noncorresponding MAX effector AVR-PikD.

We further tested whether RGA5[HMA3] and RGA5[HMA4] could trigger RGA4-dependent cell death upon recognizing AVR-PikD in the *N. benthamiana* leaves and rice protoplasts. As shown in Fig. 2c, d, and e, the cell death was not visible in either *Agrobacterium tumefaciens*-mediated coexpression of AVR-PikD and RGA4 with RGA5[HMA3] or with RGA5[HMA4] in *N. benthamiana* or cotransfection in the rice protoplasts. As a control, RGA4-dependent cell death was induced by the combination of AVR-Pia with RGA5 both in the *N. benthamiana* leaves and in the rice protoplasts (Fig. 2c, e). These results suggested that the binding of AVR-PikD by RGA5[HMA3] or RGA5[HMA4] is insufficient to trigger RGA4-dependent cell death, thus requiring further optimization of residues implicated in the receptor-activation step.

### RGA5[HMA5] carrying the mutations of RGA5[HMA3] and RGA5[HMA4] can trigger RGA4-dependent plant cell death upon the perception of AVR-PikD

Previous studies have revealed that RGA5-HMA recognizes AVR1-CO39 with a distinct interface from that in Pik1-HMA for binding to AvrPikD[10,13] and that RGA4 is derepressed to activate plant cell death upon the recognition of AVR-Pia by RGA5[8,14]. Since RGA5[HMA3] and RGA5[HMA4] carried a single engineered interface for binding to AVR-PikD

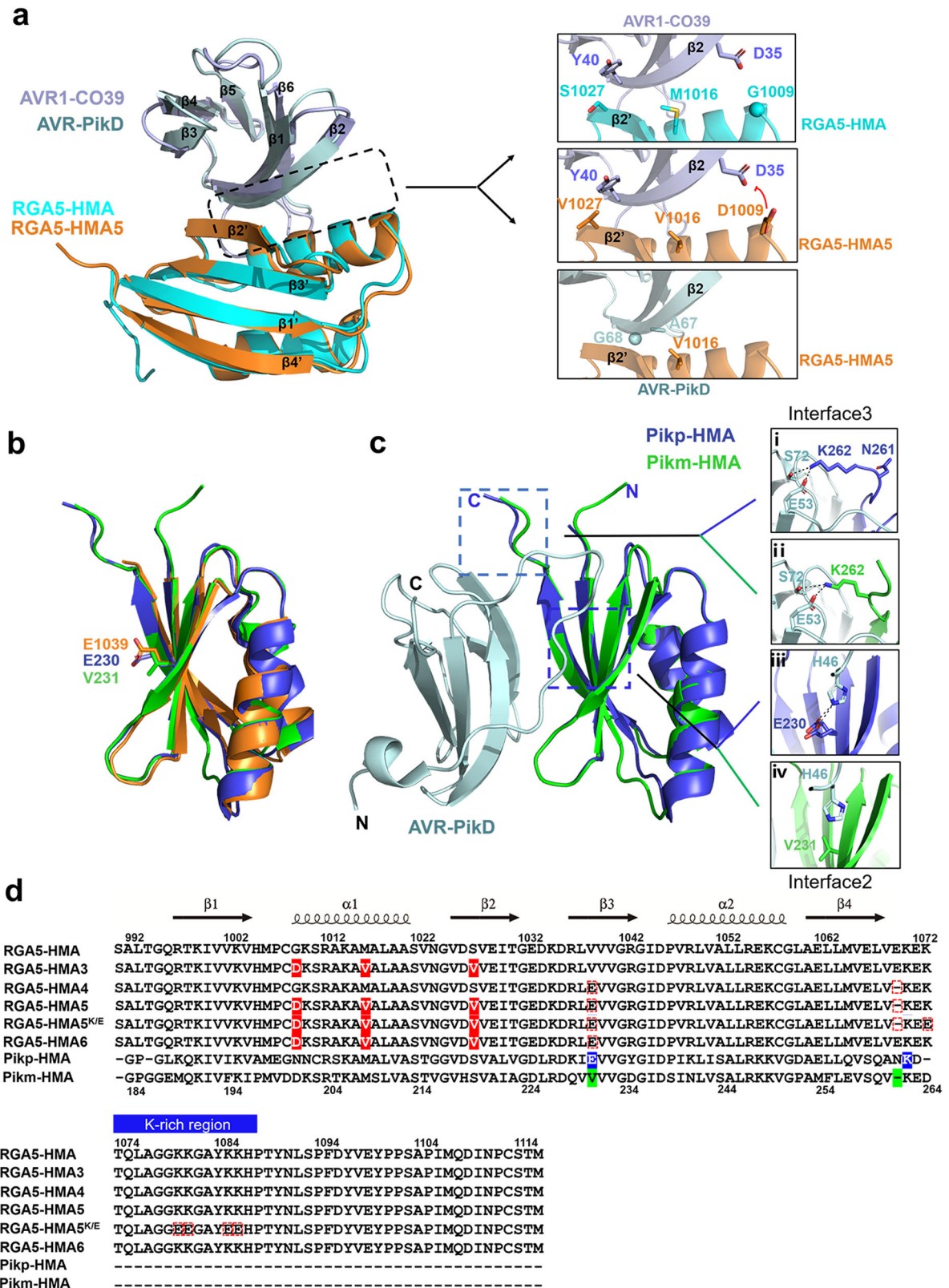

**Fig. 1 | Designing RGA5-HMA mutants capable of recognizing the non-corresponding effector AVR-PikD. a** The residues involved in the binding interface in the complexes of AVR1-CO39/RGA5-HMA and AVR-PikD/ RGA5-HMA5 were predicted by the structural superposition of the HMAs and the effectors. **b** The structural superposition of Pikp1- (PDB: 6G10), Pikm1- (PDB: 6FU9) and RGA5-HMA5 HMA domains. E1039 in RGA5-HMA5 labeled in orange corresponds to E230 in Pikp (blue) and V231 in Pikm (green) HMAs. **c** The two AVR-PikD-binding interfaces in Pikp1 or Pikm1 HMA domain. **d** Amino acid sequence alignment of RGA5-HMA mutants with RGA5-HMA and Pikp/Pikm-HMA. Labeled in the red box are modified residues in the RGA5-HMA mutants corresponding to the residues in Pikp1/Pikm HMA that were labeled in blue and green. Mutated residues in RGA5-HMA blocking the AVR-Pia binding are indicated in red shade. In RGA5-HMA mutants, E1070 was deleted, and V1039 was mutated into E, respectively. Secondary structural features of the HMA domains are shown above the alignment.

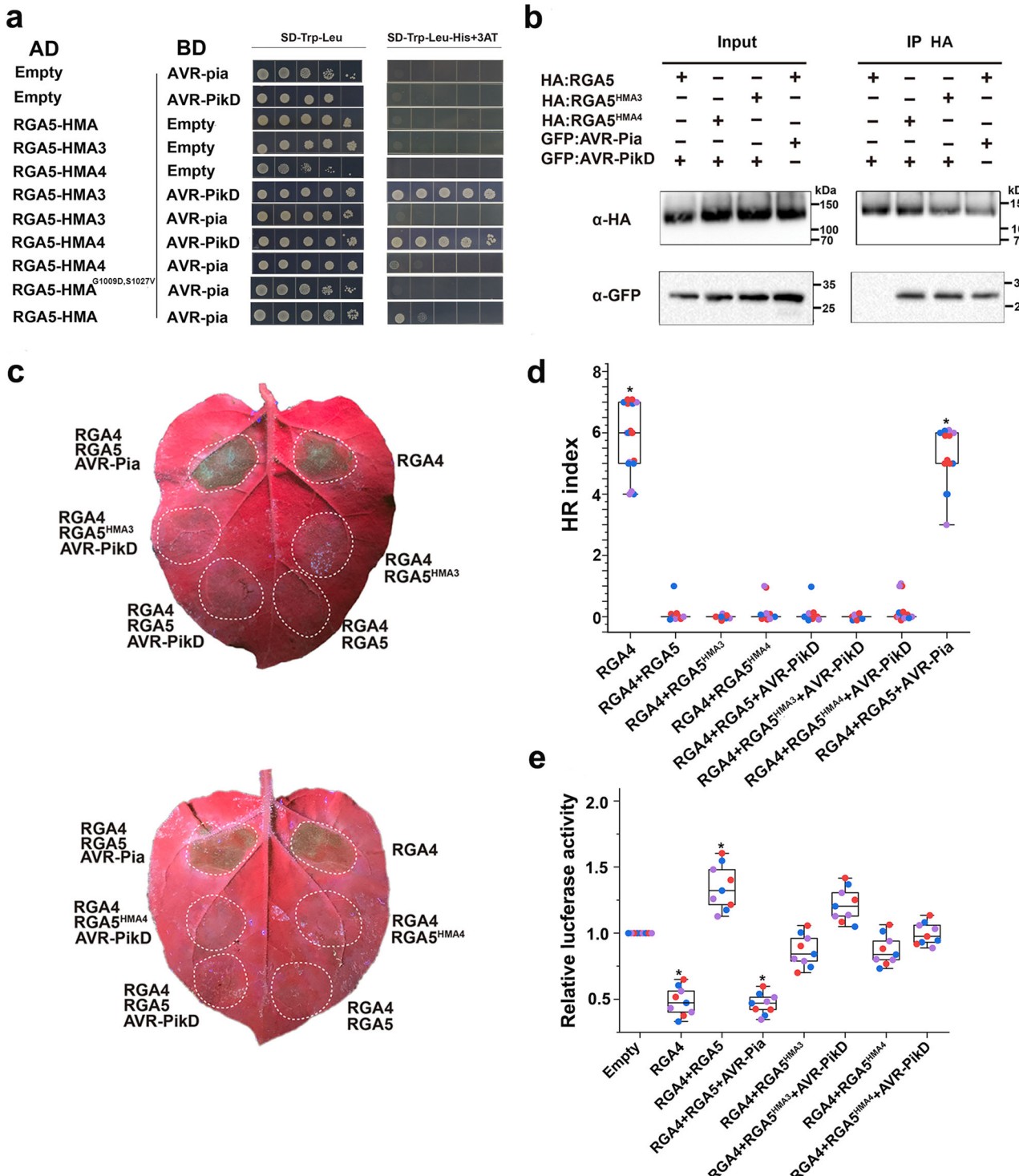

**Fig. 2 | Functional analysis of RGA5^HMA3 and RGA5^HMA4 in the *N. benthamiana* leaves and rice protoplasts.** The interaction of RGA5-HMA3 or RGA5-HMA4 with AVR-PikD was verified by Y2H (**a**) and Co-IP in *N. benthamiana* (**b**). The experiment was repeated thrice with similar results. **c** Images of the *N. benthamiana* leaves coinfiltrated with RGA5^HMA3 or RGA5^HMA4, and RGA4, AVR-PikD and silencing suppressor p19. Leaves at three days post infiltrations were photographed under the UV light. **d** HR index of different combinations in representative pictures (**c**) was scored based on representative pictures for different values of HR indices[17]. Twenty

biological replicates were used in each group. Three independent groups in different colors were labeled in box plots. Differences among the samples were assessed by Tukey's HSD test ($p < 0.01$). **e** The LUC activity in rice protoplasts cotransfected with different vector combinations. RGA4 was set as the positive control, and empty vectors served as the negative control. Significant differences with empty vector samples are labeled with an asterisk and assessed by Dunnett's HSD test ($p < 0.05$). The assays were repeated three independent times. The three distinct colors represent three independent experiments.

and failed to trigger RGA4-mediated cell death, we reasoned that a functional designer RGA5 might require both an effector-binding interface and RGA4 derepression motifs within the HMA domain and its adjacent regions. Therefore, we created RGA5-HMA5 by combining the mutations present in RGA5-HMA3 and RGA5-HMA4. Meanwhile, we determined the crystal structure of RGA5-HMA5 and confirmed it was similar to that of the wild-type RGA5-HMA (RMSD = 0.6 Å) (Fig. 1b and Supplementary Fig. 1). Y2H and MBP pull-down assays showed that RGA5-HMA5 specifically interacted with AVR-PikD but not with AVR-Pia, while RGA5-HMA interacted with AVR-Pia but not with AVR-PikD (Fig. 3a, b). Microscale thermophoresis (MST) analyses revealed that RGA5-HMA5 binds to AVR-PikD with a dissociation constant ($Kd$) of 12 μM, while RGA5-HMA binds to AVR-Pia with a $Kd$ of 35 μM (Fig. 3c), indicating that RGA5-HMA5 has a higher binding affinity for AVR-PikD. Further Co-IP assays indicated that all RGA5-HMA, HMA3, HMA4 and HMA5 interacted with AVR-PikD but not with RGA4 with or without AVR-PikD in *N. benthamiana* (Supplementary Fig. 2). We then generated RGA5[HMA5] by replacing the wild-type RGA5-HMA with RGA5-HMA5, and performed Co-IP in *N. benthamiana* by transiently coexpressing HA-RGA5[HMA5]/GFP-AVR-Pia, HA-RGA5[HMA5]/GFP-AVR-PikD, HA-RGA5/GFP-AVR-Pia or HA-RGA5/GFP-AVR-PikD. As shown in Fig. 3d, HA-RGA5[HMA5] but not HA-RGA5 was coimmunoprecipitated with GFP-AVR-PikD. These results indicate that RGA5-HMA5 can specifically interact with AVR-PikD in vivo and in vitro (Supplementary Table 1).

To verify whether RGA5[HMA5] is able to activate RGA4-dependent plant cell death upon recognizing AVR-PikD, the RGA4/RGA5[HMA5] pair was first transiently coexpressed with AVR-PikD in the *N. benthamiana* leaves. As shown in Fig. 3e, the coexpression induced cell death similar to RGA4/RGA5 with AVR-Pia. In contrast, cell death was not visible when RGA4/RGA5 with AVR-PikD, RGA4/RGA5[HMA5] with AVR-Pia, RGA4/RGA5, RGA4/RGA5[HMA5], AVR-Pia or AVR-PikD were infiltrated in the *N. benthamiana* leaves (Fig. 3e) although the proteins were expressed in the leaves (Supplementary Fig. 3). We further measured the luciferase reporter activity in rice protoplasts after expressing the different combinations of proteins. As shown in Fig. 3f, the coexpression of RGA5[HMA5]/RGA4 with AVR-PikD significantly reduced the luciferase reporter activity, similar to that of RGA5/RGA4 with AVR-Pia. However, such a reduction in the luciferase reporter activity was not detected by the coexpression of RGA4/RGA5 with AVR-PikD, and RGA4/RGA5[HMA5] with or without AVR-Pia (Fig. 3f), and RGA4/RGA5m1 or RGA4/RGA5m1m2 with AVR-PikD (Supplementary Fig. 4 and Supplementary Fig. 5). In addition, the expression of RGA5[HMA5] or AVR-PikD alone in rice protoplasts could not cause an apparent immune response, indicating that RGA5[HMA5] had no cell death-inducing activity but still retained the ability to repress the activity of the RGA4-induced cell death (Fig. 3f). Altogether, these results indicate that RGA5[HMA5] gained the AVR-PikD recognition specificity *in planta*, thereby inducing RGA4-dependent cell death, consistent with the observations in the *N. benthamiana* leaves (Supplementary Table 1).

### Transgenic rice expressing RGA5[HMA5] and RGA4 confers complete resistance to the blast fungus expressing AVR-PikD

To test the function of the three designer RGA5 mutants in rice, *Oryza sativa* cv. Nipponbare protoplasts were transfected with a combination of NLRs and AvrPikD along with luciferase, which showed that the expression of RGA4 or RGA4/RGA5/AVR-Pia and RGA4/RGA5[HMA5]/AVR-PikD led to a significant reduction in luciferase reporter activity, as compared with other combinations, namely RGA5, RGA5[HMA5], AVR-PikD, RGA4/RGA5, RGA4/RGA5[HMA3], RGA4/RGA5[HMA5], RGA4/RGA5[HMA3]/AVR-PikD and RGA4/RGA5[HMA4]/AVR-PikD (Figs. 2e and 3f). The above results indicate that only RGA5[HMA5] but not RGA5[HMA3] and RGA5[HMA4] could cause RGA4-mediated cell death in rice protoplasts upon AVR-PikD recognition. We thus generated five independent transgenic rice lines by co-transforming *RGA4* and *RGA5[HMA5]* into Nipponbare, a rice cultivar lacking *Pia* and *Pik*, and tested whether their T1 generation

lines resist infection by the *M. oryzae* strains expressing *AVR-PikD*. The T1 lines of *RGA4/RGA5*[17], two monogenic lines IRBLa (expressing only *Pia*) and IRBLk (expressing only *Pik*) of Lijiangxintuanheigu (LTH)[22], and Nipponbare were used as controls. The *M. oryzae* wild-type strain DG7 lacking functional *AVR-Pia* and *AVR-PikD* and its transformants expressing *AVR-Pia* or *AVR-PikD* were used to infect the control and transgenic lines. As expected, the monogenic lines IRBLa and IRBLk were resistant to the DG7 transformants expressing *AVR-Pia* and *AVR-PikD*, respectively, while Nipponbare was susceptible to DG7 and its transformants expressing *AVR-Pia* or *AVR-PikD* (Fig. 4), confirming that the *M. oryzae* strains used in the assay were reliable. We then inoculated the transgenic rice lines expressing *RGA4/RGA5[HMAS]* or *RGA4/RGA5* by wound inoculation[17]. As shown in Fig. 4, small resistant lesions were formed in the *RGA4/RGA5[HMAS]* transgenic lines after inoculation with the DG7 transformants expressing *AVR-PikD* but not with the wild-type strain DG7 or the transformants thereof carrying *AVR-Pia* (Fig. 4a, b). Meanwhile, the transgenic rice lines expressing *RGA4/RGA5* were resistant only to the transformants expressing *AVR-Pia*, but not to DG7 or the transformants expressing *AVR-PikD* (Fig. 4a, b). We further estimated *in planta* biomass of *M. oryzae* in the inoculated rice lines, verifying that the DG7 transformants expressing *AVR-PikD* were significantly limited in proliferation in the *RGA4/RGA5[HMAS]* transgenic lines and IRBLk but not in the other rice lines (Fig. 4c). In addition, qPCR analysis confirmed that the NLR gene pairs and the effector genes were correctly expressed in the transgenic rice lines and *M. oryzae* during infection (Supplementary Fig. 6). Altogether, the above results demonstrated that the designer NLR receptor gene *RGA5[HMAS]* coexpressed with *RGA4* in transgenic rice plants could confer specific resistance to the *M. oryzae* strains expressing *AVR-PikD*, mimicking Pik/AVR-PikD-mediated resistance.

### RGA5[HMA5] harbors multiple AVR-PikD-binding interfaces, including the lysine-rich stretch tailing the HMA ID

To understand how RGA5[HMA5] triggers RGA4-dependent plant cell death, we identified peptides bound by AVR-PikD at the C-terminus of RGA5[HMA5] (982-1,116 aa), including the RGA5-HMA5 domain, by the hydrogen/deuterium exchange coupled with mass spectrometry (HDX-MS). In total, 17 peptides were identified, covering 80% of the RGA5[HMA5] C-terminus (Supplementary Fig. 7). The difference in deuterium uptake between the RGA5-HMA5 C-terminus and the complex of RGA5-HMA5 C-terminus with AVR-PikD was recorded at 60, 300, 600 and 1,200 s. As shown in Fig. 5a and Supplementary Fig. 7a, six peptides (1010–1018, 1021–1026, 1030–1039, 1055–1060, 1068–1095, 1109–1116 aa) of RGA5-HMA domain were decreased of 1.3, 0.56, 0.8, 0.88, 1.74 and 0.66 deuterons, respectively. As expected, two peptides (1010–1018 aa and 1021–1026 aa) located within the interface of RGA5-HMA binding to AVR1-CO39 and Pik-HMA/AVR-Pia, and one peptide (1030–1039 aa) corresponded to the interface of Pik1-HMA binding to AVR-PikD (Fig. 1a, b)[13,21,23]. In the interface corresponding to the Pik1-HMA interface binding to AVR-Pik, E1039 of RGA5-HMA5 forms salt bridges with the side chains of H46 of AVR-PikD (Fig. 1c). Notably, two peptides (1068–1095 aa and 1109–1116 aa) identified were from the C-terminal Lys-rich tail immediately following RGA5-HMA5 (Fig. 5a). Within this interface, the forward shift of K1070 resulted from the front E deletion in RGA5-HMA5 was designed to mimic K262 of Pikm-HMA (Fig. 1c), which interacts with E53 of AVR-PikD by a salt bridge[21]. To verify the significance of K1070 and V1039E in derepressing the RGA5[HMA5]/RGA4 complex, we generated the E53A and H46A mutations within AVR-PikD, and the mutants of RGA5-HMA6 and RGA5[HMA6], in which E1070 was kept as in RGA5. E53A but not H46A mutation in AVR-PikD abolished the interaction with RGA5-HMA5 (Fig. 5b). To our surprise, RGA5[HMA6] failed to trigger RGA4-dependent cell death by AVR-PikD, although RGA5-HMA6 retained AVR-PikD binding capability (Fig. 5b, c), suggesting that the E1070 deletion with the forward shift of K1070 is crucial to the capability of RGA5-HMA5 to derepress RGA4 for

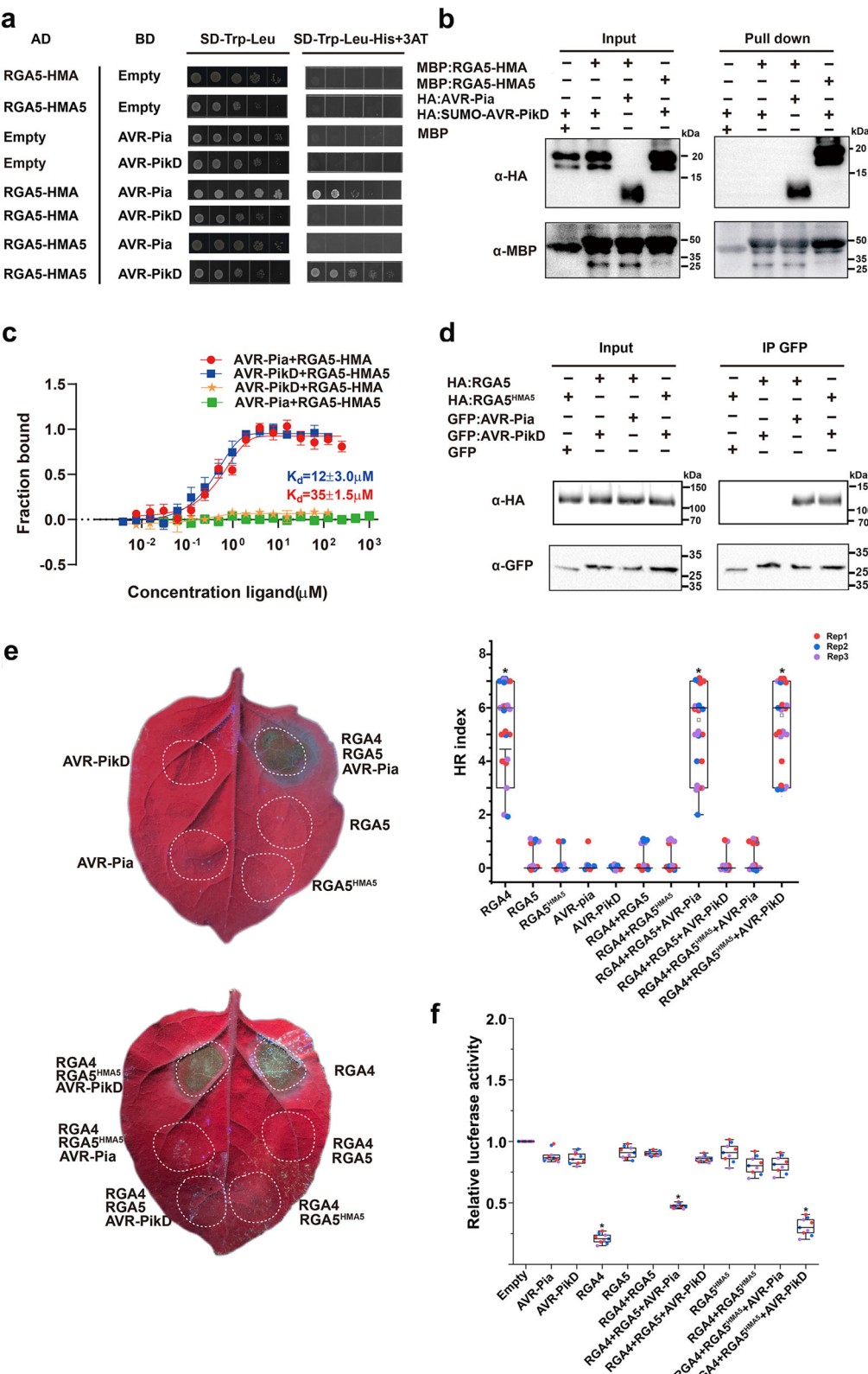

inducing the rice immunity. Previous studies reported that, in contrast to other MAX effectors, AVR-PikD has an N-terminal negatively charged patch consisting of Asp and Glu residues[10] (Fig. 5a and Supplementary Fig. 8). Y2H assays showed that AVR-PikD without the N-terminal Loop (named AVR-PikD$^{\Delta N}$) failed to interact with Pikm-HMA, RGA5-HMA5, RGA5-HMA3, and RGA5-HMA4 (Fig. 5b). In addition, the N-terminal Loop alone was unable to interact with the C-terminal tail of RGA5-HMA5 (Fig. 5b). Furthermore, as mentioned above, two peptides bound by AVR-PikD were located within the C-terminal Lys-rich stretch tailing the HMA ID, which is absent from Pik1-HMA[9,17]. In a previous study, we showed that substituting all the positively charged Lys residues with negatively charged Glu residues in the C-tail is essential to the interaction of the designer receptor RGA5$^{HMA2}$ with the noncorresponding AVR-Pib. We thus generated

**Fig. 3 | The designer NLR receptor RGA5$^{HMA5}$ activates RGA4-dependent plant cell death upon recognizing the noncorresponding MAX effector AVR-PikD. a** Y2H assays show the interaction of RGA5-HMA5 with AVR-PikD but not with AVR-Pia. **b** Pull-down assays show specific interaction of HA-AVR-PikD with MBP-RGA5-HMA5 but not with MBP-RGA5-HMA. HA-AVR-PikD, HA-AVR-Pia, MBP-RGA5-HMA5, and MBP-RGA5-HMA proteins were individually expressed in *E. coli*. Fusion proteins in different combinations were visualized by immunoblotting with the anti-HA and anti-MBP antibodies. The experiment was repeated thrice with similar results. **c** MST analyses show the dissociation constants ($Kd$) between the RGA5-HMA or RGA5-HMA5 domain and the effectors. The $Kd$ were calculated and error bars represent the means ± SD from data from three independent biological replicates. **d** Coimmunoprecipitation assays showing the interaction of RGA5$^{HMA5}$ with AVR-PikD in *N.benthamiana*. Co-IP proteins were detected by using anti-HA and anti-GFP antibodies, respectively. The experiment was repeated thrice with similar results.

**e** Representative leaf images showing specific cell death in *N. benthamiana* leaves after coinfiltration of the *A. tumefaciens* strains carrying RGA5$^{HMA5}$ (fused with HA) with RGA4 (fused with Flag) and AVR-PikD (fused with GFP), or RGA5 (fused with HA) with RGA4 (fused with Flag) and AVR-Pia or RGA4. Images were taken three days after the infiltrations under the UV light. HR index was scored based on representative pictures as previously reported[18]. Twenty biological replicates were used in each group. Three independent groups in different colors were labeled in box plots. Differences among the samples were assessed by Tukey's HSD test ($p < 0.01$). **f** The LUC activity in rice protoplasts after transfection with different vector combinations. RGA4 was set as the positive control, and the empty vectors were served as the negative control. Significant differences with empty vector samples are labeled with an asterisk and assessed by Dunnett's HSD test ($p < 0.01$). The assays were repeated three times with similar results.

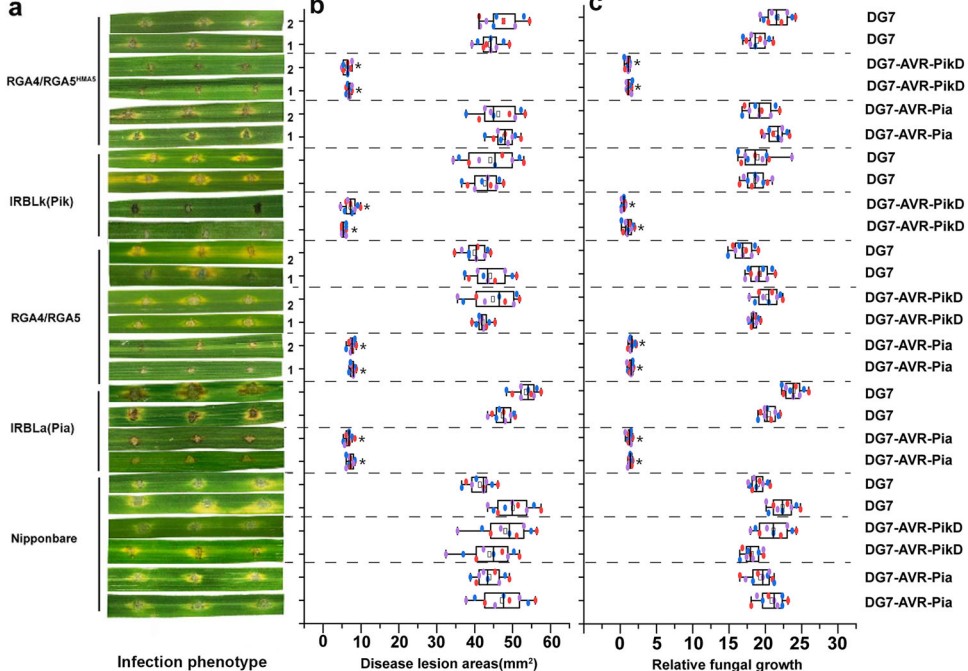

**Fig. 4 | The transgenic rice lines expressing *RGA5$^{HMA5}$*/*RGA4* confer specific resistance to the *M. oryzae* strains expressing the noncorresponding effector AVR-PikD. a** Images showing disease reactions (resistant or susceptible) on the leaves of Nipponbare and its transgenic lines expressing *RGA4/RGA5$^{HMA5}$* or *RGA4/RGA5*, and the LTH monogenic lines IRBLk (expressing *Pik*) and IRBLa (expressing *Pia*) following the inoculation with the *M. oryzae* DG7 strain expressing AVR-PikD or AVR-Pia. The wild-type strain DG7 and its transformant expressing *AVR-PikD* or *AVR-Pia* caused susceptible reactions on Nipponbare. The infection assays were performed in triplicate. Conidial suspensions of *M. oryzae* for all the inoculations were adjusted at a concentration of 10⁵/ml. Representative photos of the inoculated

leaves from two independent lines were taken four days after inoculation. **b** Box-and-whisker plots showing statistics on the sizes of lesions formed as described in **a**. Lesion areas were measured by ImageJ. **c** Bar graphs showing the biomass of *M. oryzae* in the infected rice leaves as described in **a**. The fungal biomass was quantified by measuring the expression levels of *MoPot*2 in relation to the rice ubiquitin gene. Values are means with standard deviations of nine independent biological replications from three independent rice lines. Significance analysis compared to Nipponbare is labeled with an asterisk and performed with Student's *t*-test ($p < 0.05$).

RGA5-HMA5$^{K/E}$, an RGA5-HMA5 mutant, by replacing all the Lys residues with the Glu residues except K1070 in the C-terminal tail located immediately after RGA5-HMA5. As shown in Fig. 5b, c, RGA5$^{HMA5K/E}$ could interact with AVR-PikD but lost the capability to trigger RGA4-dependent cell death in rice protoplasts by AVR-PikD, suggesting that the C-terminal positively charged Lys residues are crucial to the capability of RGA5$^{HMA5}$ to derepress RGA4 for inducing the rice immunity. In addition, RGA5-HMA5 had a peptide bound by AVR-PikD that was located at the α2 helix-loop5 (Fig. 5a). Altogether, the above results revealed that RGA5$^{HMA5}$ harbors at least three interfaces for the interaction with AVR-PikD, including an interface within the C-terminal Lys-rich tail, in which the positively charged residues are crucial for RGA5$^{HMA5}$ to derepress RGA4 for triggering the rice cell death. Interestingly, RGA5 orthologs in different accessions of rice have more

mutations in the HMA domain and C-tail than in the other domains, and their C-tail-based phylogenetic tree is similar to their HMA-based phylogenetic tree, suggesting that the C-tail seems to be co-evolved with the HMA domain (Supplementary Fig. 9f, g). In addition, some non-integrated HMA proteins in rice also have such a C-terminal tail, which is highly diversified (Supplementary Fig. 9h).

## Discussion

Structure-guided rational engineering of NLRs is emerging as a promising approach to altering their recognition spectra and specificities[24]. Several proof-of-concept studies have been recently reported on engineering NLRs and their IDs. The first designer NLR receptor was Pikp-1$^{NK-KE}$, which was generated by engineering the Pikp-1 HMA ID, showing an expanded recognition spectrum against related

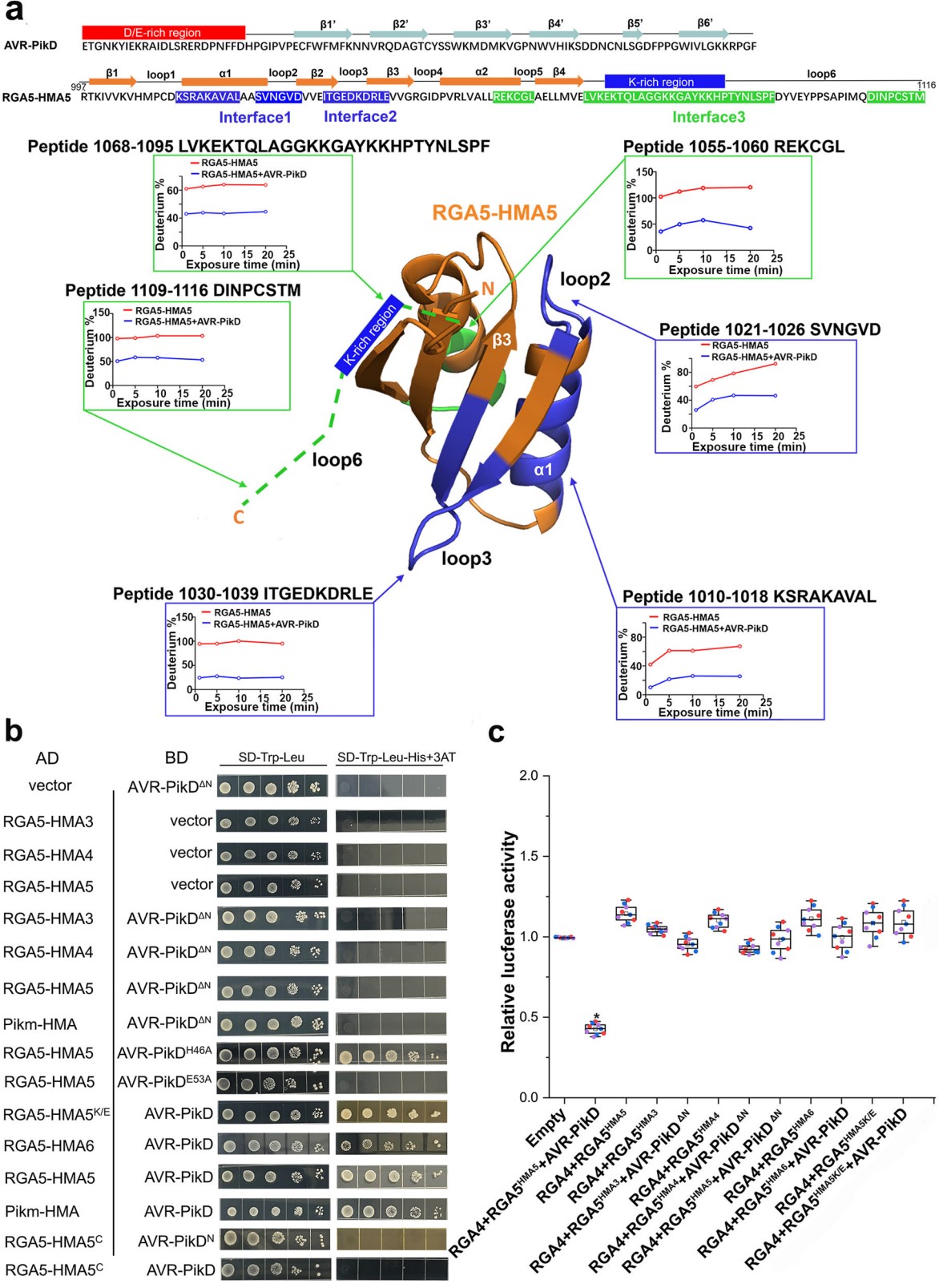

AVR-Pik alleles[16]. Kourelis et al. contrived a series of Pikm-1 mutants called Pikobodies by replacing the Pikm1 HMA ID with nanobodies of fluorescent proteins, which could recognize antigen fluorescent proteins and trigger immunity in *N. benthamiana*[25]. Tamborski *et al.* created a mutant of wheat NLR receptor Sr33 capable of recognizing the avirulence effector AvrSr50 from stem rust by switching amino acid residues in the LRR domain with the AvrSr50-binding residues in

Sr50[26]. However, the functionality of these designer NLRs remains to be verified in their host plants. We recently reported a designer NLR named RGA5[HMA2] that conferred complete resistance in rice to the *M. oryzae* expressing the noncorresponding avirulence effector AVR-Pib[17]. RGA5[HMA2] is the first example of designer NLRs conferring complete resistance in plants[24]. Here, we report yet another designer rice NLR receptor named RGA5[HMA5], which conferred complete resistance in

**Fig. 5 | AVR-PikD-binding Interfaces of RGA5^HMA5 identified by the epitope mapping based on hydrogen/deuterium exchange mass spectrometry (HDX-MS). a** The C-terminus of RGA5^HMA5 (997-1116 aa) was used to map peptides bound by AVR-PikD. The interaction interfaces are colored in blue and green on the top, and the complex model of RGA5-HMA5 bound to AVR-PikD is shown at the bottom. The green dashed lines indicate the C-terminal Lys-rich region after RGA5-HMA5. The graphs showed the deuterium percentage of HDX data over 20 min labeling for the C-terminal fragments of RGA5^HMA5 corresponding to secondary structures in the absence (red) or presence (blue) of AVR-PikD. A significant reduction in the values of deuterium percentage indicates less exchange and more opportunities for binding to AVR-PikD. The assays were performed three independent times. **b** Y2H assays show that the N-terminal loop of AVR-PikD is crucial to the interaction between the HMA domains and the effector. AVR-PikD^ΔN and RGA5-HMA5^K/E indicate the AVR-PikD mutant lacking the N-terminal loop and the RGA5-HMA5 mutant carrying the Glu substitutions of Lys residues in the C-terminal tail after the HMA domain, respectively. RGA5-HMA5^C, the C-tail fragment after the HMA domain of RGA5^HMA5; AVR-PikD^N, the N-terminal loop fragment of AVR-PikD. The others were as described in Fig. 1. **c** A bar graph shows that the LUC activity was significantly reduced in rice protoplasts after transfection with the RGA5^HMA5/RGA4/AVR-PikD vector combination but not either by the RGA5^HMA6/RGA4/AVR-PikD or by the RGA5^HMA5K/E/RGA4/AVR-PikD vector combination. Significant differences with empty vector samples are labeled with an asterisk and assessed by Dunnett's HSD test ($p < 0.05$). The assays were repeated three independent times.

transgenic rice plants to the *M. oryzae* strains expressing the non-corresponding effector AVR-PikD. More importantly, we delimit and resurface the interfaces within and outside the HMA ID of RGA5, which not only impart de novo effector-binding but also activate the NLR-mediated resistance. Our studies demonstrate that rationally engineering the HMA ID in RGA5 can generate a series of designer NLR receptors with distinct resistance profiles[17], which will be useful for efficiently breeding multiline cultivars to maintain the durability of NLR-mediated resistance[20,27].

Wang et al. created a variant of RRS1 by adding a *Phytoplasma* effector SAP05-dependent degron domain to the C-terminus of RRS1, which elicited the hypersensitive response (HR) in *N. tabacum* leaves triggered by the effector but failed to confer full resistance in transgenic *Arabidopsis* to the *Phytoplasma* infection[19]. By engineering RGA5-HMA in accordance with Pikp1-HMA, Cesari *et al.* generated two designer NLR receptors, which shared the engineered β2β3β4 interface of RGA5-HMA and gained a high binding affinity to the noncorresponding effector AVR-PikD in addition to the corresponding effector AVR-Pia but were unable to confer specific blast resistance in transgenic rice plants against *M. oryzae* strains expressing AVR-PikD[18]. Similarly, RGA5^HMA3, RGA5^HMA4 and RGA5^HMA6 generated in this study, which carried a single or two engineered interfaces, also failed to trigger RGA4-dependent rice immunity. Compared to these designer RGA5 receptors unable to trigger RGA4-dependent rice immunity, RGA5^HMA5 contains two engineered interfaces within RGA5-HMA and one interface in the Lys-rich stretch tailing the HMA ID (Fig. 5), indicating that synthetic sensor NLR receptors may require multiple engineered interfaces to be active in host plants. Notably, we showed that replacing the five Lys residues with Glu residues in the C-terminal Lys-rich tail abolished the capability of RGA5^HMA5 to induce RGA4-dependent cell death, indicating that the adjacent C-terminal tail functions not only as an interface for binding to AVR-PikD but also plays a regulatory role for RGA5^HMA5 to derepress RGA4. Regulation of IDs by their adjacent domains may not be unique to RGA5. A previous study showed that DOM 4 and DOM 6 adjacent to the WRKY ID in RRS1 contribute to autoinhibition and activation of RRS1/RPS4 immune receptor complex[28]. Therefore, concurrent modification of IDs and their adjacent domains rather than ID alone may be a prerequisite to creating designer sensor NLR receptors. In addition, we showed that some RGA5 orthologs in different accessions of rice have more mutations in the C tail and some rice non-integrated HMA proteins also have a similar but highly divergent C tail to that in RGA5 (Supplementary Fig. 9). These similar but highly divergent C tails in RGA5 orthologs and non-integrated HMA proteins may be useful for designing new RGA5 receptors.

We previously reported that RGA5^HMA2 gained specific resistance to the *M. oryzae* strains expressing the noncorresponding MAX effector AVR-Pib but lost the inherent resistance to the *M. oryzae* strains expressing the corresponding MAX effector AVR-Pia[17]. Here again, RGA5^HMA5 lost resistance to the *M. oryzae* strains with AVR-Pia, although it gained the capability to resist infection by the *M. oryzae* strains expressing the noncorresponding MAX effector AVR-PikD. In contrast,

RGA5m1 and RGA5m1m2 retained the resistance of RGA5 to the *M. oryzae* strain carrying the corresponding AVR-Pia but did not gain the capability to resist infection by the *M. oryzae* strains carrying the noncorresponding effector AVR-PikD[18]. These studies raise a key question of whether molecular engineering of IDs in sensor NLR receptors can confer an expanded spectrum of resistance to pathogens expressing unrelated effectors. However, Maidment *et al.* recently generated two variants of Pikp-1 with expanded blast resistance profiles in transgenic rice plants[29]. Therefore, further investigation is required to optimize reported designer RGA5 receptors and determine the relationship between the interfaces and key residues thereof for binding to the corresponding and noncorresponding effectors.

*N. benthamiana* or *N. tabacum* has been widely adopted as a convenient heterologous system to assay plant immune responses, such as HR. However, recent studies showed that designer NLR receptors that gained the capability to recognize effectors and trigger HR in *N. benthamiana* or *N. tabacum* may not always enable complete resistance in host plants[18,19]. These studies alert us that using the heterologous overexpression system to assess the synthetic NLRs seems insufficient. As such, there is a need for a homologous transient expression system to guide us on NLR gain-of-function engineering. We showed that RGA5-based designer NLRs were able to trigger RGA4-dependent cell death in rice protoplasts and confer complete blast resistance in transgenic rice, indicating that the cell death in rice protoplasts induced by RGA5-based designer NLRs is consistent with the intact plant blast resistance. Therefore, using homologous system-based bioassays prior to making transgenic plants may be a prerequisite to determining whether an engineered NLR gains recognition capacity to confer resistance.

In summary, we created a designer NLR receptor RGA5^HMA5 that confers full resistance in transgenic rice plants to the blast fungus *M. oryzae* strains expressing the noncorresponding effector AVR-PikD. More importantly, we show that synthetic sensor NLR receptors require concurrent structure-guided engineering of multiple interfaces within and outside IDs. Notably, we found that the C-terminal lysine-rich stretch tailing the HMA ID in RGA5^HMA5 is an interface crucial to both recognizing the MAX effectors and activating RGA4-dependent rice immunity. We also suggest that utilizing homologous systems is important to assay designer NLR receptors. This study not only represents a significant advance toward structure-guided rational engineering of NLR receptors but also has implications for designing sensor NLRs by engineering their IDs.

## Methods
### Generation of constructs
*RGA5-HMA* (nucleotides 2944 to 3348) were obtained by gene synthesis (Genecreate, Wuhan). Point mutations were generated by using the Quik Change Site-Directed Mutagenesis Kit (TransGen Biotech, Beijing). For the *E. coli* protein expression, RGA5-HMA and RGA5-HMA5 with HA-MBP-tag were independently ligated to the pETMBP1a vector, and AVR-PikD with HA-sumo-tag and AVR-Pia with HA-tag ligated to

pETSUMO1a and pHAT₂, respectively. For the Y2H assay, the *RGA5-HMA* mutants and the effector genes (*AVR-Pia* and *AVR-PikD*) were separately cloned into pGADT7 and pGBKT7 (Clontech, Palo Alto). For the transient expression assays in *N. benthamiana*, RGA5, RGA5^HMA5, and RGA4 fused with HA or Flag, and AVR-Pia and AVR-PikD effectors with GFP tag were individually cloned into the pCAMBIA 1305 vector carrying the 35S promoter. For the assays using rice protoplasts, *RGA4*, *RGA5*, *RGA5^HMA5*, *RGA5^HMA6*, *RGA5^HMASK/E*, *LUC*, *AVR-Pia*, and *AVR-PikD* were inserted into pUC19, which also drives the expression through the 35S promoter. For rice transformation, *RGA5* and *RGA5^HMA5*, and *RGA*4 under their native promoters were separately inserted into the vectors pCAMBIA1305 and pCAMBIA1300, respectively. For *M. oryzae* transformation, *AVR-Pia* and *AVR-PikD* with their native promoters were, respectively, cloned into pKN. The primers used to amplify the abovementioned genes are listed in Supplementary Table 1. Datasets for full-length sequences of all the constructs are available in the Zenodo repository (https://doi.org/10.5281/zenodo.8133144).

### *M. oryzae* strains and rice transformation
The wild-type strain of *M. oryzae* DG7 and its transformants carrying *AVR-Pia* or *AVR-PikD* were routinely maintained on oatmeal tomato agar (OTA) plates[30]. For *M. oryzae* transformation, the strain DG7 was grown in complete medium (CM) for two days, and protoplasts were isolated from the CM-grown mycelia were used for transformation with linearized pKN plasmids carrying *AVR-Pia* or *AVR-PikD*. The resulting transformants were screened with geneticin at 400 μg/ml (Invitrogen, Carlsbad)[31]. Conidia harvested from OTA plates at 25 °C were used in infection assays. Rice transformation was conducted by using the *Agrobacterium tumefacien*–mediated method[32]. For the transformation, constructs pCAMBIA1305:RGA5, pCAMBIA1305:RGA5^HMA5, and pCAMBIA1300:RGA4 were separately introduced into the *A. tumefaciens* strain EHA105 by electroporation, and the resulting *A. tumefaciens* transformants carrying pCAMBIA1305:RGA5 or pCAMBIA1305:RGA5^HMA5 with pCAMBIA1300:RGA4 were grown on LB medium at 28 °C and co-transformed into the rice cultivar Nipponbare, which lacks *Pia* and *Pik*.

### Infection assays of the *M. oryzae* strains on rice lines
Conidia of *M. oryzae* DG7 and its transformants were adjusted at a concentration of $10^5$ conidia per ml with sterilized water containing 0.025% Tween-20 and wound-inoculated on the leaves of Nipponbare, two Lijiangxintuanheigu monogenic lines IRBLa (with *Pia*) and IRBLk (with *Pik*), and the transgenic rice lines. The inoculated leaves were incubated at 26 °C in the dark for 36 h and then in a 12-h light/12-h dark cycle for an additional 4 d[17]. Disease lesions formed on the leaves were scored four days post-inoculation and measured with ImageJ (https://imagej.net/). The infection assay of rice lines by *M. oryzae* was repeated three times at least.

### qPCR assays
Total RNA was extracted from the *M. oryzae*-infected rice leaves using the RNA extraction kit (Vazyme, Nanjing) and then reverse transcribed into cDNA with the HiScript 1st Strand cDNA Synthesis Kit (Vazyme, Nanjing). qPCR was performed by the ABI Quantstudio 6 Flex PCR program (Thermo-Fisher Scientific, Waltham), with the actin genes in *M. oryzae* and rice as internal controls for normalizing the gene expressions. The differences in the expression levels were measured between the wild-type and transgenic lines or DG7 and its transformants expressing effectors. The assays were performed with three independent replicates. Primers used for the assays were listed in Supplementary Table S2.

### Yeast two-hybrid (Y2H) assays
*AVR-PikD* and *AVR-Pia* without the signal peptide sequence were cloned into the plasmid pGBKT7 as bait vectors, whereas *RGA5* and its

HMA domain mutants into pGADT7 as prey vectors. The yeast strain AH109 was co-transformed with the prey and corresponding bait vectors following the protocol provided by the Yeastmaker™ yeast transformation system (Clontech, Palo Alto). The yeast transformants were grown on SD/-Trp/-Leu medium, and the interactions were assessed on SD/-Trp/-Leu/-His plates with X-α-gal.

### Cell death assays in *N. benthamiana* leaves and rice protoplasts
The transient protein expression in *N. benthamiana* was used as a heterologous system to assay plant cell death triggered by AVR-Pia or AVR-PikD along with RGA4 and RGA5 or its mutants[33]. The *A. tumefaciens* GV3101 strains containing the different constructs were cultured in LB (Luria-Bertani) medium containing 20 μg/ml rifampicin and 100 μg/ml kanamycin. For each of the four GV3101 strains containing *P19*, *RGA4*, *RGA5^HMA* or *RGA5^HMA3/4/5*, *AvrPia* or *AVR-PikD*, OD₆₀₀ was adjusted to 0.5 and mixed at final OD₆₀₀ of 2.0. The strain mixtures were incubated for 3 h at room temperature and infiltrated into the 3-week-old *N. benthamiana* leaves[17]. After 48 h incubation in the dark, cell death around the infiltration sites was scored and photographed under the UV light[34].

Luciferase activity in rice protoplasts was used as an indicator of host cell death triggered by RGA4 and RGA5 or its mutants along with AVR-Pia or AVR-PikD. The activity was measured by using the luciferase assay system (Promega, Madison), which was performed 16 h after transfection with the plasmid combinations containing AVR-Pia or AVR-PikD along with RGA4 and RGA5 or its mutants. Rice protoplasts were prepared from Nipponbare leaves[33]. All peeled and cut leaves in protoplast isolation buffer were wrapped by aluminum foil and incubated for 3 h at room temperature in the dark with shaking at 60 rpm. Leaf tissue was filtered by nylon cell strainer, and protoplasts were collected at $100 \times g$ for 3 min. The plasmid combinations mixed with the empty and 5 μg LUC plasmids to total 20 μg in 20 μl (LUC:NLR:AVR = 5 μg:5 μg:5 μg) were cotransfected into rice protoplasts via the poly-ethylene glycol method[35]. The assay was repeated in three independent experiments.

### Co-IP and immunoblotting
The *N. benthamiana* leaves infiltrated with different GV3101 strains as described in the cell death assay were ground into powder with liquid nitrogen and then homogenized with the extraction buffer (25 mM Tris–HCl pH 7.5, 1 mM EDTA, 150 mM NaCl, 10% glycerol, 2% poly-vinylpolypyrolidone [PVPP], 5 mM DTT, 1x protease inhibitor, and 1 mM PMSF). The homogenates were centrifuged at 13,800 RCF for 10 min, and the supernatant was then applied to anti-GFP agarose beads (50 μl) at 4 °C for 3 h. The beads were washed three times with the washing buffer (25 mM Tris–HCL pH 7.5, 1 mM EDTA, 150 mM NaCl, 10% glycerol, 5 mM DTT, and 1x protease inhibitor). The proteins were separated by 10% SDS-PAGE gels and detected by immunoblotting with the first anti-GFP, anti-FLAG, or HA-tag antibodies (Sigma, Poole) and the second anti-rabbit or anti-mouse IgG-peroxidase antibody (Sigma, Poole). Membranes were washed three times with TBST buffer (20 mM Tris, pH 8.0, 150 mM NaCl, 0.05% Tween-20), and the immunoblot signals were visualized using the HRP chemiluminescent substrate (Millipore, Billerica). Rubisco small submit stained by Ponceau S was used to verify equal protein loading as the control.

### Interaction assays by pull-down and microscale thermophoresis
For the pull-down assay, MBP-RGA5-HMA5, MBP-RGA5-HMA, HA-AVR-Pia, and HA-SUMO-AVR-PikD proteins were expressed in *E. coli* BL21 (DE3) at 289 K for 10 h by adding 0.4 mm IPTG and 50 μg/ml kanamycin. Recombinant proteins were purified by Ni-Chelating Sepharose Fast Flow column and Superdex 75 10/300GL column (Cytiva, Vancouver) equilibrated in binding buffer (20 mM Tris, 150 mM NaCl, 5 mM DTT, 4 mM EDTA, pH 7.4, 1% Triton X-100). Recombinant RGA5-HMA, RGA5-HMA5 with the effector proteins were mixed and applied

for Anti-MBP beads in a binding buffer (50 μl), incubated with gentle rotation for 3 h at 4 °C. Then the resin was washed five times with the binding buffer and boiled for 10 min. All the proteins were loaded onto 10% SDS-PAGE gel for separation and then transferred onto the PVDF membrane (Millipore), and subsequently detected with the anti-MBP antibody (ABclonal, Wuhan).

For the MST assay, RGA5-HMA and RGA5-HMA5 were separately labeled with the fluorescent dye NT-647 from kit MO-L001 of Nano Temper. All the labeled proteins were dissolved in the buffer containing 20 mM PBS, 150 mM NaCl, 0.05 % (v/v) Tween-20, pH 7.4, and mixed with different concentrations of effectors (AVR-PikD or AVR-Pia). Finally, the Kd (the dissociation constant) values between the effector and the RGA5 HMAs were measured using Monolith NT.115 (NanoTemper Technologies) with 30% LED power and fitted with Nano Temper Analysis Software (Version 1.5.41). The assays were repeated in three independent experiments.

## Crystallization, data collection, and structure determination of RGA5-HMA5

For protein crystallization, RGA5-HMA5 (residues 982-1,116) was cloned into the vector pHAT₂ and expressed in *E. coli* BL21 (DE3). The transformed DE3 cells were grown in LB medium at 310 K to OD600 of 0.6–0.8 containing 50 μg/ml ampicillin, and expression of the recombinant protein was induced by adding 0.4 mm IPTG and further incubation at 289 K for 10 h. The cells were harvested by centrifugation and resuspended in lysis buffer (20 mm Tris–HCl, 500 mm NaCl, 20 mm imidazole, pH 7.5), followed by sonication lysis. Supernatants from the cell lysates were applied to a Ni-Chelating Sepharose Fast Flow column after washing with lysis buffer, His-tagged proteins were eluted with elution buffer (20 mm Tris–HCl, 150 mm NaCl, 500 mm imidazole pH 7.5). Proteins were then purified by gel filtration chromatography on a Superdex 75 10/300GL column (Cytiva, Vancouver) equilibrated in storage buffer (20 mm Tris–HCl, 150 mm NaCl, pH 7.5, 2 mM DTT). After gel filtration chromatography, purified proteins were concentrated to 7 mg/ml and flash-frozen and stored at −80 °C[36]. Crystals of the RGA5 HMA mutant were produced by sitting drop vapor diffusion, which occurred after three days under the same condition as the wild-type RGA5-HMA (0.2 M ammonium nitrate, 20 % [w/v] PEG 3350). The crystals were soaked into cryoprotectant containing 20 % (v/v) glycerol and flash cooling into liquid nitrogen. X-ray diffraction data were collected at the Shanghai Synchrotron Radiation facility by beamline BL19U. Data were processed using the HKL-2000 processing package[37]. The structure was solved by molecular replacement using Phaser with 5ZNE as a search model. The final structure was obtained by rebuilding using Coot[38], and further refined using PHENIX with TLS restraints[39]. The detailed statistics on data collection and refinement are listed in Table 1.

## Hydrogen/deuterium exchange coupled with mass spectrometry (HDX-MS)

RGA5-HMA5 with and without the equal molar quantity of AVR-PikD was prepared before the labeling reactions by using RGA5-HMA5 as the control. 5 μl of RGA5-HMA5 (5 mg/ml) and the complex (5 mg/ml) of RGA5-HMA5 and AVR-PikD were diluted 10-fold in the deuterium labeling buffer (20 mM PBS, pH 7.4, 150 mM NaCl, 99.8% D2O) and incubated at 25 °C. The labeling reactions were stopped at special time points (1, 5, 10, and 20 min) by adding 50 μl of ice-cold quenching buffer (4 M guanidine hydrochloride, 200 mM citric acid, and 100 mM TCEP, pH 1.8). After adding 5 μl pepsin solution (1 μM) to the reactions for 3 min digestion, the quenched samples were applied to Thermo-Dionex Ultimate 3000 HPLC system autosampler. For LC-MS/MS analysis, the peptides were separated using a 40-min linear gradient acetonitrile-water (8 to 50% containing 0.1% formic acid) at a flow rate of 125 μl/min on a reverse-phase column (Acquity UPLC BEH C18 column,1.7 μm, 2.1*50 mm, Waters, UK) with Thermo-Dionex

### Table 1 | Data collection and refinement statistics

|  | RGA5-HMA5 BL19u1 |
|---|---|
| Data collection |  |
| Wavelength (Å) | 0.9786 |
| Resolution range (Å) | 29.54–2.80 (2.90–2.80)[a] |
| Space group | P 43 21 2 |
|  | 66.00 66.00 132.14 |
| Unit cell | 90.00 90.00 90.00 |
| Total reflections | 165,102 |
| Unique reflections | 7671 (736) |
| Multiplicity | 22.6 (13.3) |
| Completeness (%) | 99.52 (98.92) |
| Mean I/sigma(I) | 24.4 (2.4) |
| **Refinement** |  |
| Wilson B-factor | 80.96 |
| Rmerge[b] | 0.19 (1.08) |
| Rmeas | 0.20 (1.11) |
| CC1/2 | 0.97 (0.89) |
| R-work[c] | 0.21 (0.36) |
| R-free | 0.25 (0.42) |
| Number of non-hydrogen atoms | 1102 |
| Macromolecules | 1102 |
| Protein residues | 146 |
| **RMSD** |  |
| Bond lengths (Å) | 0.009 |
| Bond angle (°) | 1.30 |
| **Ramachandran plot (%)[d]** |  |
| Ramachandran favored | 95.77 |
| Ramachandran allowed | 4,23 |
| Ramachandran outliers | 0.00 |
| Rotamer outliers | 0.00 |
| Clashscore | 10.84 |
| Average B-factor | 91.94 |
| macromolecules | 91.94 |
| Number of TLS groups | 11 |

[a]Numbers in parenthesis are for the highest resolution data shell.
[b]$R_{merge} = \sum_{hkl}\sum_i(|I_i(hkl) - I(hkl)\rangle|)/\sum_{hkl}\sum_i I_i(hkl)$.
[c]$R_{work} = \sum_{hkl}(||F_{obs}| - |F_{calc}||)/\sum_{hkl}|F_{obs}|$.
[d]As evaluated by MolProbity.

Ultimate 3000 HPLC system connected to a Thermo Scientific Q Exactive mass spectrometer. Mobile phase A consisted of 0.1% formic acid, and mobile phase B consisted of 100% acetonitrile containing 0.1% formic acid. The Q Exactive mass spectrometer was running in the data-dependent acquisition mode with Xcalibur 2.0.0.0 software and there was a single full-scan mass spectrum in the orbitrap (350–2000 *m/z*, 70,000 resolution). The mass spectrometer was operated at a source temperature of 250 °C and a spray voltage of 3.0 kV. Peptides were identified using an in-house Proteome Discoverer (Version PD1.4, Thermo-Fisher Scientific, USA). The search criteria were as follows: no enzyme was required; two missed cleavage was allowed; precursor ion mass tolerances were set at 20 ppm for all MS acquired in an orbitrap mass analyzer; and the fragment ion mass tolerance was set at 0.02 Da for all MS2 spectra acquired. The peptide false discovery rate (FDR) was calculated using the Percolator provided by PD. The deuterium exchange levels were determined by subtracting the centroid mass of the undeuterated peptide from the centroid mass of the deuterated peptide using HD Examiner (Version PD1.4, Thermo-Fisher Scientific, USA). The assay was performed with three technical replicates.

## Reporting summary

Further information on research design is available in the Nature Portfolio Reporting Summary linked to this article.

## Data availability

The coordinates and structure factors have been deposited in the Protein Data Bank with accession code 7DVG. Additional data supporting the findings of this study are available from the corresponding authors upon request. Source data are provided in this paper.

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

## Acknowledgements

This work was supported by grants from the Natural Science Foundation of China (Grant No. 32030089, 32293244 and 32102160), the China Agricultural Research System (CARS-01-44), the IRT Program (Grant No. IRT1042), the Pinduoduo-China Agricultural University Research Fund (Grant No. PC2023A01005), and the National Key Research and Development Program of China for Young Scientists (2022YFD1401400). The authors thank BL17U1 at the Shanghai Synchrotron Research Facility (SSRF) beamline and BL19U1 at the National Facility for Protein Science in Shanghai, Zhangjiang Laboratory, China, for providing technical support and assistance in data collection and analysis. We also thank Sheng Yang He at Duke University for the critical reading of the manuscript and Xiaolin Tian at Tsinghua University Branch of China National Center for Protein Sciences Beijing for technical help in the HDX-MS experiment.

## Author contributions

Y.L.P. and J.L. designed research; X.Z., Y.L., G.Y., S.W., T.Z., X.W., M.M., and L.G. performed research; Y.L.P., J.L., X.Z., Y.L, G.Y., T.Z., X.W., M.M., D.W., L.G., V.B., and H.G. analyzed data; and Y.L.P., J.L., X.Z., V.B. H.G., and Y.L. wrote the paper.

## Competing interests

The authors declare no competing interests.
