## [Peer Review File · Nature Communications]

The synthetic NLR RGA5HMA5 requires multiple interfaces within and outside the integrated domain for effector recognitionReviewer #1 (Remarks to the Author):

In this elegant NLR engineering study, authors rationally interchange several effector binding surfaces and uncover novel contribution of lysine rich C-terminal lysine rich 'tail' in RGA5 to the activation of the immune signaling through helper NLR, RGA4. The study combines genetics and biochemistry to dissect effector/receptor interactions and utilizes several orthogonal assays, including transient expression in *Nicotiana benthamiana*, rice protoplasts and transgenics to demonstrate effectiveness of synthetic NLR. Overall, I feel that the work is very polished already and deserves publication. All of the presented results are already well supported. My comments are mostly about data presentation, availability of reagents that should be required and additional evolutionary analyses that could inform the reader and put results uncovered in the specific receptor in a larger NLR biodiversity context.

Specific comments on data presentation and availability of reagents presented in the paper:

- For all figures please use box plots in place of bar graphs and include all individual data points overlaid on the graph.
- For protoplast assays specifically, please, indicate if presented data is one experiment (if so, how many technical replicates) or an aggregate of all three experiments. Please, see Saur et al 2019 (<https://doi.org/10.1186/s13007-019-0502-0>) for an example of more transparent data presentation that I am looking for.
- Please, include full length sequences of all synthetic constructs used in the study as supplemental information/dataset. Plasmid maps, if available, can be uploaded to Zenodo, and plasmids to Addgene where applicable. Please, include how these resources would be available to the community otherwise.
- Include seed stock availability information.
- Figure 4 (especially images of leaves) are very difficult to see. I would suggest to provide a larger version of the images/graphs.

Additional evolutionary analyses that would enhance this work:

- Currently, the phylogenetic distribution of lysine rich tail which is one of the most exciting discoveries presented in this work is unclear. Is it presented in any other RGA5 orthologs? Is it present as an untranslated sequence past their stop codons? Is it present in any non-integrated HMA proteins? I think a simple blast search both in protein space but also in protein against genomic DNA space can reveal the origin of this functionally important region.
- A more complex analysis would be to test co-evolution of the C-terminal tail with the rest of RGA5 protein (are engineered surfaces and the lysine rich region show signature of co-evolution?) and with RGA4. I would understand if authors do not have expertise in this area and although this would greatly enhance the work, I don't see it as a requirement for publication.

Reviewer #2 (Remarks to the Author):

The manuscript by Zhang et al. engineers four RGA5-HMA mutants based on the structure information and finds out one of the mutants confers complete resistance in transgenic rice plants to the *M. oryzae* strains expressing the noncorresponding effector AVR-PikD. The authors identify three interfaces bound by AVR-PikD in the RGA5HMA5 through HDX-MS and biochemistry assays. This work demonstrates that RGA5HMA5 requires multiple surfaces within and outside the IDs to recognize noncorresponding effectors and activate helper NLR-mediated resistance.

Overall, this work provides a significant advance toward structure-guided designing of sensor NLRs with distinct specificity of effectors for directional crop breeding and also raises the complex effector-binding and receptor-activation mechanisms behind NLRs. I would make the following suggestions to strengthen the current version of the manuscript.

Major notes

1. The authors conclude that some key residues in RGA5-HMA5 are crucial for derepressing RGA4.

However, all the binding assays are performed without helper NLRs, only between RGA5 and effectors, which undermines the points about the HMA engineering effects on RGA4. Césari et al. report the RGA5 CC domain is required but not sufficient for repression of RGA4-mediated cell death, while the RATX1 domain (HMA domain) is dispensable (Césari et al., *The EMBO Journal*, 2014). In this manuscript, the authors find that RGA5-HMA5 can trigger RGA4-dependent plant cell death upon the perception of AVR-PikD. Does it mean the engineered HMA domain of RGA5 can interact and repress RGA4? Test whether RGA5-HMA/HMA3/HMA4/HMA5 can bind RGA4 with or without AVR-PikD would improve strongly on this point. The failure to trigger RGA4-dependent cell death may be caused by specific residues combination which affords the correct formation of RGA5-RGA4 heterodimer.

2. Even though both Pik1-HMA/RGA5-HMA and AVR-PikD/AVR-COR39 are structurally similar, the interface of Pik1/AVR-PikD and RGA5/AVR-Pia or AVR1-COR39 are different ($\beta 2$ - $\beta 3$ - $\beta 4$ sheet and $\alpha 1$ helix- $\beta 2$ interface respectively), and this binding pattern difference may be related to the correct function of corresponding helper NLRs. So, engineering the residues in the RGA5/AVR-Pia or AVR1-COR39 for the RGA5/AVR-PikD interaction may bring in the complexity of results, e.g., RGA5-HMA3/HMA4 can bind to AVR-PikD but failed to trigger RGA4-dependent cell death. However, this interface difference may be used for engineering the receptor that can interact with both types of effectors. It will be exciting if the structure of RGA5-HMA with AVR-PikD or even with RGA4 can be solved.

3. The three interfaces the authors identified are localized in four opposite parts of RGA-HMA5. Based on the structure of AVR-PikD, it is hard to imagine that AVR-PikD can interact with RGA-HMA all around at all sides.

4. The authors should provide the peptide coverage and the independent sample number of RGA5-HMA5 in the HDX experiment. For HDX experiments, we can quickly figure out the regions that contribute to the interaction, but the authors should be cautious when claiming the role of a particular residue. The exchange rate of any overlapping adjacent peptides should be considered.

5. The peptide (1030-1039 aa) in loop three identified in the HDX experiment, which corresponded to the interface of Pik1-HMA binding to AVR-PikD, is also participating in the homodimer formation of RGA5-HMA (as shown in RGA5-HMA structure and Xi et al., *Mol Plant Pathol.* 2022). Is it possible that the different exchange rates of this peptide may come from the disruption of AVR-PikD on the formation of the homodimer of RGA5-HMA?

6. Do any binding test results demonstrate that the decreasing binding affinity of Pikm1-HMA is caused by the "looping out" residue N261 in Pikp-HMA? Does the E1070 in RGA5-HMA have the same effects? Because Pikp-HMA can bind to AVR-PikD, and RGA5HMA5K/E can bind AVR-PikD but lose the capability to trigger RGA4, it looks like the key factor is the interference from the opposite charge of E1069 to K1070 in RGA5-HMA, not steric-hindrance effect. This can be further proved by the Pikp-1NK-KE, which also loses response to AVR-PikD.

7. The HDX experiment results show that the two peptides in C-ter tail of HMA5 change from highly flexible to structured when the AVR-PikD is added, and this tail is crucial for derepressing RGA4 for triggering the rice cell death. Can this tail bind to AVR-PikD?

Minor notes

1. In figure 1d, adding residue numbers in the alignment sequences may make the reader easier to find out corresponding residues.

2. It's better to show the sequence of substituted residues of RGA5-HMA5K/E in Fig. 1

3. The contents of figure 2e are not elucidated clearly in the main text, which is mentioned in line 207, which refers to "The above results" in line 205,

4. In line 118, the PDB code of AVR-PikD and the complex of AVR1-CO39/RGA5-HMA used for structural superimposition should be provided.

Reviewer #3 (Remarks to the Author):

In this manuscript the authors successfully engineered paired NLR RGA4/RGA5a to recognize AVR-PikD which normally is not recognized by wildtype RGA4/RGA5a. The new RGA5^{HMA5} is capable of binding AVR-PikD and then derepress RGA4 to induce

defense to the blast fungus expressing AVR-PikD. To achieve this the authors had to engineer three interfaces including the one outside integrated domain.

Several studies demonstrating synthetic sensor NLRs with extended or altered effector recognition specificities have been already published including the one from the same group (<https://doi.org/10.1073/pnas.2110751118>) and others (<https://doi.org/10.7554/eLife.47713>). Also, natural resistance to AVR-PikD do exist which lessens the novelty of this manuscript. However, in this manuscript the authors have demonstrated that an interface outside of the integrated domain was crucial for the correct functioning of RGA4 which is a novel finding and may have broader impact in the future NLR engineering studies.

The authors should seriously consider generating transgenic Nipponbare expressing RGA4 with RGA5a, RGA5^{HMA2} and RGA5^{HMA5} altogether and test it against individual rice blast fungus strains which express AVR-Pia, AVR-Pib or AVR-PikD. Provided that such transgenic plant exhibit full resistance against all rice blast strains tested, it would mean that synthesizing and stacking novel resistance is a real possibility and thus have much broader impact.

Overall, the authors have provided enough evidence to support their claims although the details of experiments are sometimes lacking due to very limited methods section. For example, on page 15, the authors have stated that they have constructed *N. benthamiana* expression clones using pCAMBIA1305 and rice protoplast assay clones using pUC19 without identifying which promoters were used to drive the expression. I believe pCAMBIA1305 has 35S promoter but pUC19 does not have any plant promoters. Which promoter was used for rice protoplast luciferase assays? Also, qPCR assays do not describe how the comparisons were done.

Again, the results provided by the authors appears to be high quality and adequate to support their claims. However, the methods section is too brief to be reproduced stand-alone. The authors should utilize supplemental information section to describe their experimental methods in detail even if the methods used were previously published elsewhere.

Line 1: The title is too broad. The authors should change "sensor NLR receptors" to "sensor NLR RGA5^{HMA5}", since the authors do not know if that statement is also true in Pik-1/Pik-2-based engineered NLRs.

Lines 22-23: Not all sensor NLRs have integrated domains nor are they paired NLRs.

Line 34: should remove the word "noncorresponding"

Line 39: there is no such thing as "gene-for-gene diseases". Should change to "crop diseases".

Line 40: The reference #1 is inadequate. Should change to <https://doi.org/10.1038/s41559-018-0793-y>

Line 50: "However, sensor NLRs usually carry" – this is true only in paired NLRs. ZAR1 is a sensor NLR which do not require any helper NLRs and do not contain integrated domain. Should change to "In paired NLRs, sensor NLRs usually"

Line 57: references missing for Pik1 and RGA5 molecular engineering.

Line 57: "in rice to confer" should change to "in rice that confer"

Line 83: "designer sensor NLRs to trigger immune" should change to "designer sensor NLRs to trigger full immune"

Lines 378-379: "Quik Change Site-Directed Mutagenesis Kit (Transgen)" I could not find company Transgen.

Line 421: "yeast strain Y2H AH109" should change to "yeast strain AH109" AH109 should not be in italic font

Line 443: GV3101 should not be in italic font

Line 445: Tris-HCL should be Tris-HCl

Line 447: "The homogenates were centrifuged" should describe the exact centrifugation settings

Line 456: "ponceau" should be "Ponceau S"

Responses to the reviewers' comments

Reviewer #1: In this elegant NLR engineering study, authors rationally interchange several effector binding surfaces and uncover novel contribution of lysine rich C-terminal lysine rich 'tail' in RGA5 to the activation of the immune signaling through helper NLR, RGA4. The study combines genetics and biochemistry to dissect effector/receptor interactions and utilizes several orthogonal assays, including transient expression in *Nicotiana benthamiana*, rice protoplasts and transgenics to demonstrate effectiveness of synthetic NLR. Overall, I feel that the works is very polished already and deserves publication. All of the presented results are already well supported. My comments are mostly about data presentation, availability of reagents that should be required and additional evolutionary analyses that could inform the reader and put results uncovered in the specific receptor in a larger NLR biodiversity context.

Response: We genuinely appreciate all of your valuable comments and suggestions. We have addressed all the concerns and suggestions in the revised version of the manuscript and made amendments as per suggestions. Please see the tracked changes in the revised version of the manuscript.

Major suggestions/concerns:

1. For all figures please use box plots in place of bar graphs and include all individual data points overlaid on the graph.

Response: Thank you for the suggestion! As per your suggestion, we have replaced the bar graphs with the box plots with overlaid data points obtained from three biological replications, which are color-coded. The changes were made in all the figures.

2. For protoplast assays specifically, please, indicate if presented data is one experiments (if so, how many technical replicates) or an aggregate of all three experiments. Please, see Saur et al 2019 (<https://doi.org/10.1186/s13007-019-0502-0>) for an example of more transparent data presentation that I am looking for.

Response: The data presented was obtained from three independent experiments, with three technical replicates per experiment. The three experiments are coded by distinct colors in the box plots.

3. Please, include full length sequences of all synthetic constructs used in the study as supplemental information/dataset. Plasmid maps, if available, can be uploaded to Zenodo, and plasmids to Addgene where applicable. Please, include how these resources would be available to the community otherwise.

Response: We have made a supplementary dataset for full-length sequences of all the constructs, which has been uploaded to Zenodo (<https://doi.org/10.5281/zenodo.8133144>).

4. Include seed stock availability information.

Response: We generated five independent transgenic lines expressing RGA5^{HMA5} with RGA4, which are stocked in our lab, but, at present, there are only a few seeds for these lines. However, we are propagating the seeds, which will be available to the scientific community upon direct request to the corresponding authors.

5. Figure 4 (especially images of leaves) are very difficult to see. I would suggest to provide a larger version of the images/graphs.

Response: We do have high-resolution TIFF images included in Figure 4, and will furnish these TIFF images.

6. Currently, the phylogenetic distribution of lysine rich tail which is one of the most exciting discoveries presented in this work is unclear. Is it presented in any other RGA5 orthologs? Is it present as an untranslated sequence past their stop codons? Is it present in any non-integrated HMA proteins? I think a simple blast search both in protein space but also in protein against genomic DNA space can reveal the origin of this functionally important region.

Response: These are excellent questions! As reported in this study, the C-terminal lysine-rich tail immediately after RGA5-HMA plays a crucial role in regulating RGA4-dependent cell death. Following your suggestion, we downloaded RGA5 orthologs in seven rice accessions that have paired RGA4/RGA5 orthologs, including cultivated and wild species of rice published in a recent study (Shang et al., 2022, Cell Research 32:878–896) and made an alignment of their amino acid sequences of the RGA5 orthologs, excluding the Pias2 orthologs that have the DUF761 domain instead of the HMA domain as in RGA5, and presented the alignment in the new supplementary Fig S9a. The comparison revealed that all the RGA5 orthologs have a similar C-tail albeit evident difference between rice accessions. We also did a BLAST search of the RGA5 C-tail against the Nipponbare genome of rice and didn't find any significant hit, suggesting that the lysine-rich C tail is not present in any untranslated sequence. We generated the designer RGA5^{HMA5} by using the cDNA of RGA5 from Kitaake, confirming that the lysine-rich C tail is not an untranslated sequence past the stop codon. Nevertheless, We did pBLAST against the NCBI database and found that some non-integrated HMA proteins of rice also carry lysine-rich C-tails albeit highly diversified (Supplementary Fig S9h), suggesting that the C-terminus of RGA5 may have originated from integration of an HMA protein. Notably, a recent study reported that rice RGA4/RGA5 allelic NLRs have six distinct integrated domains at a similar position after the leucine-rich repeat domain, including Pias2 orthologs that have

DUF761 domain for recognizing AVR-Pias, and suggested that the integrated domains are originated from integration of endogenous sequences matching the domains mediated by CID (conservation and association with IDs) motif (Shimizi et al., 2022, PNAS 119, e2116896119).

7. A more complex analysis would be to test co-evolution of the C-terminal tail with the rest of RGA5 protein (are engineered surfaces and the lysine rich region show signature of co-evolution?) and with RGA4. I would understand if authors do not have expertise in this area and although this would greatly enhance the work, I don't see it as a requirement for publication.

Response: Thank you for the suggestion! To investigate the coevolution between the C-tail and the rest of the domains of RGA5, we have performed simple domain-specific phylogenetic analyses of CC, NB-ARC, LRR, HMA and C-tail domains. The analyses, as shown in Supplementary Fig S9 c to g, suggest that the variations in the C-tail of RGA5 orthologs may be coevolved with the HMA domain but not the rest of the domains.

Reviewer #2: The manuscript by Zhang et al. engineers four RGA5-HMA mutants based on the structure information and finds out one of the mutants confers complete resistance in transgenic rice plants to the *M. oryzae* strains expressing the noncorresponding effector AVR-PikD. The authors identify three interfaces bound by AVR-PikD in the RGA5HMA5 through HDX-MS and biochemistry assays. This work demonstrates that RGA5HMA5 requires multiple surfaces within and outside the IDs to recognize noncorresponding effectors and activate helper NLR-mediated resistance.

Overall, this work provides a significant advance toward structure-guided designing of sensor NLRs with distinct specificity of effectors for directional crop breeding and also raises the complex effector-binding and receptor-activation mechanisms behind NLRs. I would make the following suggestions to strengthen the current version of the manuscript.

Response: We sincerely thank the reviewer for the comments and suggestions.

Major comments

1. The authors conclude that some key residues in RGA5-HMA5 are crucial for derepressing RGA4. However, all the binding assays are performed without helper NLRs, only between RGA5 and effectors, which undermines the points about the HMA engineering effects on RGA4. Césari et al. report the RGA5 CC domain is required but not sufficient for repression of RGA4-mediated cell death, while the RATX1 domain (HMA domain) is dispensable (Césari et al., The EMBO Journal, 2014). In this manuscript, the authors find that RGA5-HMA5 can trigger RGA4-dependent plant cell death upon the perception of AVR-PikD. Does it mean the engineered HMA domain of RGA5 can interact and repress RGA4? Test whether RGA5-HMA/HMA3/HMA4/HMA5 can bind RGA4 with or without AVR-PikD would improve strongly on this point. The failure to trigger RGA4-dependent cell death may be

caused by specific residues combination which affords the correct formation of RGA5-RGA4 heterodimer.

Response:

This is a critical point. Césari et al. (2014), in their EMBO journal paper, performed the domain deletion analysis of RGA5 to delimit the role of individual domains thereof in effector-independent RGA4-mediated cell death in *Nicotiana benthamiana*. The authors concluded that the CC domain of RGA5 is crucial (required although not sufficient) for functional interaction between RGA5 and RGA4 and that the RATX1 (aka HMA) domain is not required in RGA4 repression but for AVR recognition. This conclusion doesn't exclude the regulatory role of the HMA domain of RGA5 in RGA4 derepression upon recognition of the MAX effectors. In nature, RGA4 is derepressed to trigger HR upon recognition of the MAX effector by RGA5 via its HMA. That is why, we did not investigate the impact of canonical domains within RGA5 on RGA4 repression but instead focused on the effects of resurfacing the HMA domain and the C-terminal tail following HMA to gain binding affinity with the noncorresponding effector AVR-PikD under the assumption that as long as there exists the interaction, RGA4-mediated cell death would occur. Therefore, the interaction assays between RGA5-HMA variants and the noncorresponding effector were performed without RGA4. However, we test the NLR variant RGA5^{HMA5} with and without RGA4 for bioassays (Fig.3d, e and f; Fig.5c). Like the RATX1 domain in Césari et al., we found that the engineered HMA domains alone do not repress RGA4. In addition, our results show that none of the RGA5-HMA, -HMA3, -HMA4 or -HMA5 domain interacted with RGA4 with or without AVR-PikD by co-IP assay in Fig S2. Although RGA5-HMA3 and RGA5-HMA4 could bind to AVR-PikD, RGA5-HMA5, RGA5^{HMA3} and RGA5^{HMA4} failed to trigger RGA4-dependent cell death as RGA5^{HMA5}. This discrepancy may not be caused by losing the ability to interact with RGA4. We suppose that some amino acids in both the HMA domain and the C-terminal tail may be important to derepress RGA4 by causing allosteric changes (please refer to the description on RGA5^{HMA3}, and RGA5^{HMA4}, RGA5^{HMA6} RGA5^{HMA5K/E} in the ms from lines 240-290). The structural mechanism that depresses the RGA4 activity will give the reasons for that. Our and other groups are still working to solve the complex structures of RGA4/5 with or without the effectors to understand the structural mechanism to depress the RGA4 activity.

2. Even though both Pik1-HMA/RGA5-HMA and AVR-PikD/AVR-COR39 are structurally similar, the interface of Pik-1/AVR-PikD and RGA5/AVR-Pia or AVR1-COR39 are different (β 2- β 3- β 4 sheet and α 1 helix- β 2 interface respectively), and this binding pattern difference may be related to the correct function of corresponding helper NLRs. So, engineering the residues in the RGA5/AVR-Pia or AVR1-COR39 for the RGA5/AVR-PikD interaction may bring in the complexity of results, e.g., RGA5-HMA3/HMA4 can bind to AVR-PikD but failed to trigger RGA4-dependent cell death. However, this interface difference may be used for engineering the receptor that can interact with both types of effectors. It will be exciting if the structure of RGA5-HMA with AVR-PikD or even with RGA4 can be

solved.

Response: We fully agree with the comments. The structure of RGA5-HMA with AVR-PikD or even with RGA4 will give us solid data to explain the functional activity of the mutants designed by us and the Kroj's group. Unfortunately, we have been unable to obtain the crystal of the complexes, but we are still working on it. In the meantime, we here performed HDX-MS to analyze the interfaces of RGA5-HMA with AVR-PikD. The HDX-MS results suggested that multiple interfaces of RGA5-HMA5 are involved in the interaction with AVR-PikD, and the rice protoplast assays indicate that multiple interfaces but not only single interface can confer the RGA5 recognizing AVR-PikD to activate the RGA4-dependent cell death. These indicate that both interfaces of Pik-1/AVR-PikD and RGA5/AVR-Pia or AVR1-COR39 in RGA5-HMA5 are indispensable for activating RGA4-dependent cell death. We are currently optimizing the two interfaces and the C-tail to generate a RGA5 variant that can interact with both types of effectors and confer resistance to rice blast isolates that carry either of them.

3. The three interfaces the authors identified are localized in four opposite parts of RGA-HMA5. Based on the structure of AVR-PikD, it is hard to imagine that AVR-PikD can interact with RGA-HMA all around at all sides.

Response: Thank you for the comments. RGA5-HMA5 interacts with AVR-PikD by the engineered key residues on the interface 1 (RGA5-HMA/AVR-Pia) and interface 2 (Pikp-HMA/AVR-PikD), while the C-terminal tailing of RGA5-HMA on the interface 3 as well contribute to complex formation. We reckon that RGA5-HMA5 may bind AVR-PikD in multiple modes or the flexible loop in the N-terminus of AVR-PikD extending to the three interfaces in the RGA5-HMA domain may also be important for the interaction.

4. The authors should provide the peptide coverage and the independent sample number of RGA5-HMA5 in the HDX experiment. For HDX experiments, we can quickly figure out the regions that contribute to the interaction, but the authors should be cautious when claiming the role of a particular residue. The exchange rate of any overlapping adjacent peptides should be considered.

Response: Thank you for the suggestions. We performed the HDX-MS analyses in three independent experiments and obtained similar results. The peptide coverage was added in Supplementary Fig. S7. Six peptides, including overlapping adjacent peptides in RGA5-HMA5, decreased exchange upon binding of RGA5-HMA5 with C-tail by AVR-PikD. We verified two interaction surfaces by the Y2H assays and the LUC activity in rice protoplasts.

5. The peptide (1030-1039 aa) in loop three identified in the HDX experiment, which corresponded to the interface of Pik1-HMA binding to AVR-PikD, is also participating in the homodimer formation of RGA5-HMA (as shown in RGA5-HMA structure and Xi et al., Mol Plant Pathol. 2022). Is it possible

that the different exchange rates of this peptide may come from the disruption of AVR-PikD on the formation of the homodimer of RGA5-HMA?

Response: This is a good question. The interface (RGA5-HMA3) of the RGA5-HMA homodimer contributed to the interaction of engineered RGA5-HMA with AVR-PikD based on our Y2H results (Fig 2a). So, AVR-PikD may disrupt the HMA homodimer to bind to the interface of the β 2 and α 1 in RGA5-HMA.

6. Do any binding test results demonstrate that the decreasing binding affinity of Pikm1-HMA is caused by the “looping out” residue N261 in Pikp-HMA? Does the E1070 in RGA5-HMA have the same effects? Because Pikp-HMA can bind to AVR-PikD, and RGA5HMA5K/E can bind AVR-PikD but lose the capability to trigger RGA4, it looks like the key factor is the interference from the opposite charge of E1069 to K1070 in RGA5-HMA, not steric-hindrance effect. This can be further proved by the Pikp-1NK-KE, which also loses response to AVR-PikD.

Response: Thank you for the comments. De la Concepcion et al. show that Pikm-HMA has tighter binding affinities for AVR-Pik effectors than Pikp-HMA due to the “looping out” residue N261 in Pikp-HMA (<https://doi.org/10.1038/s41477-018-0194-x>). Meanwhile, Pikp-1NK/KE, unlike RGA5-HMA that has a lysine-rich C-tail, can recognize AVR-PikD and mediate cell death in *N. benthamiana* (DOI: <https://doi.org/10.7554/eLife.47713.001>). RGA5-HMA could not bind to AVR-PikD, but E1070 deletion in RGA5^{HMA4}, resulting in the forward shift of K1071, acquired binding affinity to AVR-PikD (Fig. 2 and 3), suggesting that the forward shift of K1071 is important for binding to AVR-PikD by RGA5-HMA4 and RGA5-HMA5. We further constructed a mutant RGA5-HMA6 carrying the E1070, which was almost the same as RGA5HMA5 except keeping the E1070 and could bind to AVR-PikD. However, RGA5^{HMA6} lost the capability to activate RGA4, suggesting that the forward shift of K1071 is also required for RGA5^{HMA5} to derepress RGA4 upon the AVR-PikD recognition.

7. The HDX experiment results show that the two peptides in C-ter tail of HMA5 change from highly flexible to structured when the AVR-PikD is added, and this tail is crucial for derepressing RGA4 for triggering the rice cell death. Can this tail bind to AVR-PikD?

Response: Our HDX results indicate that some regions of the C-terminal tail are involved in the binding of RGA5-HMA5 to AVR-PikD. However, the Y2H assay could not detect the interaction between AVR-PikD and the C-terminal tail (Fig 5b). For this discrepancy, we speculate that the C-terminal tail may be involved in the binding of AVR-PikD after the AVR-Pia/AVR1-CO39 interface binds to AVR-PikD.

8. In figure 1d, adding residue numbers in the alignment sequences may make the reader easier to find out corresponding residues.

Response: We have added the residue numbers in Fig 1d.

9. It's better to show the sequence of substituted residues of RGA5-HMA5K/E in Fig. 1

Response: Revised Fig. 1 now shows the complete sequence of RGA5-HMA5.

10. The contents of figure 2e are not elucidated clearly in the main text, which is mentioned in line 207, which refers to "The above results" in line 205.

Response: Thank you! We have added the detail in the revised text to explain Fig 2e, which reads as follows: "To test the function of the three designer RGA5 mutants in rice, *Oryza sativa* cv. Nipponbare protoplasts were transfected with a combination of NLRs and AvrPikD along with luciferase, which showed that the expression of RGA4 or RGA4/RGA5/AVR-Pia and RGA4/RGA5^{HMA5}/AVR-PikD led to a significant reduction in luciferase reporter activity, compared with other combinations namely RGA5, RGA5^{HMA5}, AVR-PikD, RGA4/RGA5, RGA4/RGA5^{HMA3}, RGA4/RGA5^{HMA5}, RGA4/RGA5^{HMA3}/AVR-PikD and RGA4/RGA5^{HMA4}/AVR-PikD (Fig. 2e, 3f). The above results indicate that only RGA5^{HMA5} but not RGA5^{HMA3} and RGA5^{HMA4} could cause RGA4-mediated cell death in rice protoplasts upon AVR-PikD recognition ."

11. In line 118, the PDB code of AVR-PikD and the complex of AVR1-CO39/RGA5-HMA used for structural superimposition should be provided.

Response: Thank you! The PDB code of AVR-PikD (PDBID: 5A6W) with the complex of AVR1-CO39/RGA5-HMA (PDBID: 5ZNG) is added in line 118.

Reviewer #3: In this manuscript the authors successfully engineered paired NLR RGA4/RGA5a to recognize AVR-PikD which normally is not recognized by wildtype RGA4/RGA5a. The new RGA5^{HMA5} is capable of binding AVR-PikD and then derepress RGA4 to induce defense to the blast fungus expressing AVR-PikD. To achieve this the authors had to engineer three interfaces including the one outside integrated domain.

Several studies demonstrating synthetic sensor NLRs with extended or altered effector recognition specificities have been already published including the one from the same group (<https://doi.org/10.1073/pnas.2110751118>) and others (<https://doi.org/10.7554/eLife.47713>). Also, natural resistance to AVR-PikD do exist which lessens the novelty of this manuscript. However, in this manuscript the authors have demonstrated that an interface outside of the integrated domain was crucial for the correct functioning of RGA4 which is a novel finding and may have broader impact in the future NLR engineering studies.

Response: We are grateful to the reviewer for the comments and for acknowledging the value of our work. We have addressed all the comments in the revised manuscript and made amendments per the reviewer's suggestions.

Major comments

1. The authors should seriously consider generating transgenic Nipponbare expressing RGA4 with RGA5a, RGA5^{HMA2} and RGA5^{HMA5} altogether and test it against individual rice blast fungus strains which express AVR-Pia, AVR-Pib or AVR-PikD. Provided that such transgenic plant exhibit full resistance against all rice blast strains tested, it would mean that synthesizing and stacking novel resistance is a real possibility and thus have much broader impact.

Response: This is an excellent suggestion. Pyramiding multiple *R* genes into a cultivar is known to be an effective strategy to broaden broad-spectrum resistance. However, Pyramiding multiple *R* genes by traditional breeding is a time-consuming and laborious process. Thus, one of our goals is to broaden the anti-race spectrum by introducing a single-engineered sensor NLR. Although the present study reports that an engineered RGA5 NLR can confer resistance to the blast strains carrying AVR-PikD, and a gene with such resistance exists in nature, we are trying to generate an engineered RGA5 with extended anti-race spectrum, e.g., an engineered RGA5 NLR conferring resistance to the blast strains with both AVR-PikD and AVR-Pia or either of them. Actually, we already obtained such an engineered RGA5. Currently, we are generating transgenic Nipponbare expressing RGA4 with RGA5^{HMA5} and RGA5^{HMA2}. Generating such transgenic rice requires a long time, and we have not yet obtained T1 generation for resistance assays. Thus, we would like to include this result for comparison in the next paper, which will report an engineered RGA5 with an extended recognition profile of avirulence effectors.

2. Overall, the authors have provided enough evidence to support their claims although the details of experiments are sometimes lacking due to very limited methods section. For example, on page 15, the authors have stated that they have constructed *N. benthamiana* expression clones using pCAMBIA1305 and rice protoplast assay clones using pUC19 without identifying which promoters were used to drive the expression. I believe pCAMBIA1305 has 35S promoter but pUC19 does not have any plant promoters. Which promoter was used for rice protoplast luciferase assays? Also, qPCR assays do not describe how the comparisons were done.

Response: Thank the reviewer for this critical comment. pCAMBIA1305 does carry the 35S promoter, which was also inserted into pUC19, driving the expression of genes in rice protoplasts. Pertinent details have been furnished along with qPCR methodology and comparison in the method section of the revised manuscript.

3. Again, the results provided by the authors appears to be high quality and adequate to support their claims. However, the methods section is too brief to be reproduced stand-alone. The authors should utilize supplemental information section to describe their experimental methods in detail even if the methods used were previously published elsewhere.

Response: Thank the reviewer for this critical comment. We have supplied the method in more detail.

4. Line 1: The title is too broad. The authors should change “sensor NLR receptors” to “sensor NLR RGA5^{HMA5}”, since the authors do not know if that statement is also true in Pik-1/Pik-2-based engineered NLRs.

Response: Thank you for the suggestion! We have changed the title, which reads now as “The effector recognition by a synthetic sensor NLR RGA5^{HMA5} requires the concerted action of multiple interfaces within and outside the integrated domain”.

5. Lines 22-23: Not all sensor NLRs have integrated domains nor are they paired NLRs.

Response: that is true! Only a subset of sensor NLRs carries integrated domains and are paired with helper NLRs. We have made changes in the text to reflect this fact.

6. Line 34: should remove the word “noncorresponding”.

Response: We have removed the word “noncorresponding”.

7. Line 39: there is no such thing as “gene-for-gene diseases”. Should change to “crop diseases”.

Response: thank you! We have changed it as you suggested.

8. Line 40: The reference #1 is in adequate. Should change to <https://doi.org/10.1038/s41559-018-0793-y>

Response: Thank you! We have replaced the reference as you suggested.

Savary, S. et al. The global burden of pathogens and pests on major food crops. *Nat. Ecol. Evol.* **3**, 430-439 (2019).

9. Line 50: “However, sensor NLRs usually carry” – this is true only in paired NLRs. ZAR1 is a sensor NLR which do not require any helper NLRs and do not contain integrated domain. Should change to “In paired NLRs, sensor NLRs usually”.

Response: That is true. We have changed the text in order to avoid generalization.

10.Line 57: references missing for Pik1 and RGA5 molecular engineering.

Response: Thank you for the suggestion! We have cited references related to the molecular engineering of Pik1 and RGA5.

11. Line 57: “in rice to confer” should change to “in rice that confer”

Response: Thank you! We have changed it to “in rice that confer”.

12. Line 83: “designer sensor NLRs to trigger immune” should change to “designer sensor NLRs to trigger full immune”

Response: Thank you! We have changed it to “designer sensor NLRs to trigger full immune”.

13. Lines 378-379: “Quik Change Site-Directed Mutagenesis Kit (Transgen)” I could not find company Transgen.

Response: Thank you very much! We have added the company name (TransGen Biotech, Beijing)

14. Line 421: “yeast strain Y2H AH109” should change to “yeast strain AH109” AH109 should not be in italic font

Response: Thank you! We have corrected it to “yeast strain AH109”.

15. Line 443: GV3101 should not be in italic font

Response: Thank you! We have corrected the font style to regular.

16. Line 445: Tris-HCL should be Tris-HCl

Response: Thank you! We have corrected it to Tris-HCl.

17. Line 447: “The homogenates were centrifuged” should describe the exact centrifugation settings

Response: Thank you! We have added the exact centrifugation settings in the revised text.

18. Line 456: “ponceau” should be “Ponceau S”

Response: Thank you! We have corrected it to “Ponceau S”.

Reviewer #1 (Remarks to the Author):

The authors did a thorough revision. All of my comments have been addressed. It is great to see this excellent work polished.

Reviewer #2 (Remarks to the Author):

In the revised version, the authors provided detailed replies and additional experiments, which fully addressed my comments. In brief, this work provides a novel insight into future NLR engineering studies.

Some data presentation suggestions:

1. In lines 118 and 119, there should be a spacing between "PDBID," shown as PDB ID.
2. It looks like mistyping "TT" before $\beta 2$ in the secondary structural features of Fig. 1D.
3. In Fig. S7, using different colors to present deuterium uptake differences between the 60s and 1200s would be better.

Reviewer #3 (Remarks to the Author):

In this manuscript the authors successfully engineered paired NLR RGA4/RGA5a to recognize AVR-PikD which normally is not recognized by wildtype RGA4/RGA5a. However, several studies demonstrating synthetic sensor NLRs with extended or altered effector recognition specificities have been already published.

That is why this reviewer suggested generating a transgenic lines with original RGA4/RGA5 pair and the new RGA5HMA5 all together to see if stacking additional recognition capability is possible and thus truly expand the resistance to rice blast (rice plants resistant to rice blast strains carrying either AVR-Pia or AVR-PikD).

But the authors explain that generating a new transgenic line is time consuming and so perhaps out of scope of this manuscript.

Other concerns were mostly addressed in this revision and their findings of interface outside of the integrated domain crucial for the correct functioning of RGA4 will have broader impact in the future NLR engineering studies.

We would like to thank you and the three anonymous reviewers for the time and effort put into evaluating our manuscript entitled “The synthetic NLR RGA5^{HMA5} requires multiple interfaces within and outside the integrated domain for effector recognition”. We have now thoroughly revised the manuscript by addressing all the comments and concerns or clarifying and explaining ambiguities. As a result, the technical quality and the presentation of the manuscript have significantly improved.

We hope that this final revision and accompanying responses will be sufficient to make our manuscript suitable for publication in Nature Communications. For details, please see the appended point-by-point responses to the comments raised by the reviewers below.

Thanks,

You-Liang Peng and Junfeng Liu

RESPONSE TO REVIEWERS' COMMENTS

Reviewer #1 (Remarks to the Author):

The authors did a thorough revision. All of my comments have been addressed. It is great to see this excellent work polished.

Response: We appreciate the reviewer's constructive feedback and recognition of the importance of our work.

Reviewer #2 (Remarks to the Author):

In the revised version, the authors provided detailed replies and additional experiments, which fully addressed my comments. In brief, this work provides a novel insight into future NLR engineering studies.

Response: We are grateful to the reviewer for constructive comments and suggestions, and for acknowledging the value of our work.

Some data presentation suggestions:

1. In lines 118 and 119, there should be a spacing between "PDBID," shown as PDB ID.

Response: We have changed PDBID into PDB ID in the revised version of the manuscript.

2. It looks like mistyping "TT" before $\beta 2$ in the secondary structural features of Fig. 1D.

Response: We have deleted "TT" in Fig 1D.

3. In Fig. S7, using different colors to present deuterium uptake differences between the 60s and 1200s would be better.

Response: As per your suggestion, we have color-coded the differences in deuterium uptake between the 60s and 1200s in the revised version of the manuscript.

Reviewer #3 (Remarks to the Author):

In this manuscript the authors successfully engineered paired NLR RGA4/RGA5a to recognize AVR-PikD which normally is not recognized by wildtype RGA4/RGA5a. However, several studies demonstrating synthetic sensor NLRs with extended or altered effector recognition specificities have been already published.

That is why this reviewer suggested generating a transgenic lines with original RGA4/RGA5 pair and the new RGA5HMA5 all together to see if stacking additional recognition capability is possible and thus truly expand the resistance to rice blast (rice plants resistant to rice blast strains carrying either AVR-Pia or AVR-PikD).

But the authors explain that generating a new transgenic line is time consuming and so perhaps out of scope of this manuscript.

Other concerns were mostly addressed in this revision and their findings of interface outside of the integrated domain crucial for the correct functioning of RGA4 will have broader impact in the future NLR engineering studies.

Response: We are grateful for the reviewer's insightful feedback and suggestions, and for recognizing the significance of our research. In fact, instead of co-expressing RGA5^{HMA2} and RGA5^{HMA5}, we have already obtained an engineered RGA5 that provides resistance

to the blast strains carrying both AVR-PikD and AVR-Pia.